# Environment geometry alters subiculum boundary vector cell receptive fields in adulthood and early development

Laurenz Muessig[1], Fabio Ribeiro Rodrigues [2], Tale L. Bjerknes[3], Benjamin W. Towse [4], Caswell Barry [1], Neil Burgess [4,5], Edvard I. Moser [3], May-Britt Moser [3], Francesca Cacucci [6,7] & Thomas J. Wills [1,7] ✉

Boundaries to movement form a specific class of landmark information used for navigation: Boundary Vector Cells (BVCs) are neurons which encode an animal's location as a vector displacement from boundaries. Here we characterise the prevalence and spatial tuning of subiculum BVCs in adult and developing male rats, and investigate the relationship between BVC spatial firing and boundary geometry. BVC directional tunings align with environment walls in squares, but are uniformly distributed in circles, demonstrating that environmental geometry alters BVC receptive fields. Inserted barriers uncover both excitatory and inhibitory components to BVC receptive fields, demonstrating that inhibitory inputs contribute to BVC field formation. During postnatal development, subiculum BVCs mature slowly, contrasting with the earlier maturation of boundary-responsive cells in upstream Entorhinal Cortex. However, Subiculum and Entorhinal BVC receptive fields are altered by boundary geometry as early as tested, suggesting this is an inherent feature of the hippocampal representation of space.

Spatial cognition in the hippocampus is supported by a network of neurons tuned to an animal's position and orientation, including place[1], head direction[2] and grid[3] cells, which collectively form a cognitive map of allocentric space[4,5]. The spatial tuning of these neurons is supported by a combination of both internally-derived movement information and sensory-bound external landmarks[6–8]. One important class of external landmarks are environmental boundaries, more specifically, extended objects that form a barrier to an animal's movement[9,10]. Boundaries are thought to anchor the cognitive map at the edges of the visited environment, correcting errors accumulated in an open field[11–16].

At a behavioural level, boundary geometry serves as a strong cue for the successful retrieval of spatial memories. After disorientation, many species of animals, including human children, will search for a reward in geometrically equivalent corners of a rectangular enclosure ('spatial reorientation')[17–19]. Remembering goal locations relative to boundaries recruits the hippocampus[20], and humans remember locations close to boundaries better than those far from them[21,22].

Within the hippocampal network, an animal's location relative to boundaries is signalled by different classes of spatially-tuned neurons. Border cells, in the medial entorhinal cortex (mEC), fire when an animal is in close proximity to a boundary, usually in one specific allocentric direction[23,24]. In the subiculum, a subset of neurons shows spatial firing consistent with that described by the boundary-vector cell (BVC) model[25], which predicted the existence of BVC neurons, i.e., neurons signalling allocentric distance and direction to boundaries[10,26]. Recent

[1]Department of Cell and Developmental Biology, University College London, London WC1E 6BT, UK. [2]Institute of Behavioural Neuroscience, University College London, London WC1H 0AP, UK. [3]Kavli Institute for Systems Neuroscience, Norwegian University of Science and Technology, Trondheim 7491, Norway. [4]Institute of Cognitive Neuroscience, University College London, London WC1N 3AZ, UK. [5]UCL Queen Square Institute of Neurology, University College London, London WC1N 3BG, UK. [6]Department of Neuroscience, Physiology and Pharmacology; University College London, London WC1E 6BT, UK. [7]These authors jointly supervised this work: Francesca Cacucci, Thomas J. Wills. ✉e-mail: t.wills@ucl.ac.uk

studies suggest that signalling an organism's position relative to boundaries may be a common neural mechanism for cognitive mapping, across vertebrates[21,27,28].

Environment boundaries serve as a foundational input to the cognitive map of space during post-natal development: before grid cells emerge (before post-natal day 21), place cells are more stable and accurate when an animal occupies locations close to a boundary[29]. Border Cells, which may act as a source of stable spatial input to developing place cells, are present in the mEC from P17 onwards[30]. However, how neural responses to boundaries develop in the subiculum remains unknown. Mapping the development of the subiculum is key to understanding hippocampal development overall: the subiculum encodes location, speed, direction, axis of travel and task relevant variables[31–33], as well as position with respect to boundaries and objects[26,34]. This information is distributed to multiple brain regions including retrosplenial cortex, nucleus accumbens and anterior thalamic nuclei[35,36]. Patterns of molecular development indicate that hippocampal maturation recapitulates information flow along the tri-synaptic loop[37], predicting late maturation of the subiculum. However, to date, developing subiculum neural responses have not been studied in behaving animals.

The goals of this study were to characterise subiculum BVC properties during post-natal development, to understand the nature of the subiculum representation of boundaries, and how these change through an animal's lifespan. We categorise subiculum neurons as BVCs by fitting idealised BVC firing rate maps[25] to neural firing rate maps, obtained by recording the activity of subiculum neurons as rats explored a familiar, square-walled, open field environment. Manipulations of environment geometry revealed that the firing properties of adult BVCs departed from their canonical definition in two ways. Firstly, directional tunings are uniformly distributed in a circle, but cluster in alignment with wall orientations in a square environment, demonstrating an influence of environment geometry on BVC receptive fields. Secondly, insertion of barriers into the open field produces not only a replication of the principal field, but also an inhibition of firing on the opposite side of the barrier, indicating a role for boundary-driven inhibition in reorganising BVC firing fields.

As predicted on the basis of previous observations[37], the development of precise and stable BVC firing, as well as adult-like responses to inserted boundaries, are slower than those observed in upstream hippocampal areas. However, the fundamental geometry of the distribution of BVC receptive fields, including the influence of geometry on directional tunings in square environments, is observable at the earliest ages tested, suggesting that these are inherent features of the hippocampal coding of space.

## Results

### Detection of BVCs in adult and developing subiculum

We recorded 517 neurons from the subiculum of four adult rats and 1080 neurons from the subiculum of 17 developing (P16-P25) rats, as they explored a square, walled, open field environment (for tetrode positions see Supplemental Fig. 1). To identify BVCs in the open field, we used an exhaustive search for the best fit for each neuronal firing rate map from a set of idealised BVC firing rate maps, constructed following the model described in ref. 25. According to this model, a BVC unit fires optimally when a boundary is detected at a specific distance and allocentric direction from the animal, its receptive field taking the form of a 2-dimensional Gaussian in distance and direction (polar) space. BVC firing is not affected by the body-centred, or egocentric, bearing of the animal to the boundary (see Fig. 1A). The model parameters which were varied to create the test set of idealised BVCs were: the distance tuning, $d$; the allocentric directional tuning, $\Phi$; and the width of the distance tuning curve, $\sigma_0$ (Fig. 1A, see Methods). If the correlation between the firing rate map and the best fitting idealised BVC was greater than a threshold derived from fitting spatially-shuffled

data, the neuron was identified as a BVC (Fig. 1B). Additionally, BVCs were required to convey spatial information greater than a threshold derived from spatially-shuffled data (following[30]; see methods). Figure 1C shows representative examples of classified BVCs in adults and developing rats, with a range of goodness-of-fit from high (top) to just crossing classification threshold (bottom) illustrated for each age group.

### Characterisation of BVC spatial firing development

The number of neurons defined as BVCs was significantly greater than expected by chance, at all ages (Fig. 2A; Binomial test: $p < 0.001$, all groups. Numbers of BVCs: P16-18, 74; P19-21, 61; P22-25, 46; Adult, 187). The proportion of BVCs significantly increased between pre- and post-weaning (Z-test P19-21 vs P22-25; $Z = 2.35$, $p = 0.019$), but remained significantly lower than adults even at P22-25 (Z-test; $Z = 3.41$, $p = 0.001$). Firing rate maps (Fig. 1C) show that developing BVCs appear less spatially specific, with greater deviation from their best-fit idealised BVC. Furthermore, comparison across two consecutive trials shows decreased spatial stability in developing BVCs (Fig. 2B). The development of the spatial specificity and stability of BVCs was quantified using spatial information, BVC fit $r_{(max)}$ (see Methods), intra- and inter-trial stability (Fig. 2C–F), respectively. All scores increased with age (1-way ANOVA age: Spatial Information, $F_{(3,364)} = 16.6$, $p < 0.001$; BVC $r_{max}$, $F_{(3,364)} = 43.1$, $p < 0.001$; Inter-trial stability, $F_{(3,363)} = 74$, $p < 0.001$; Intra-trial stability, $F_{(3,364)} = 100$, $p < 0.001$). With the exception of Spatial Information, all measures increased across the pre- and post-weanling period (Tukey HSD P16-18 vs P22-25: Spatial Information $p = 0.61$, all other measures $p < 0.001$). Furthermore, all four measures remained significantly lower in developing animals, including post-weanlings, than in adult rats (Tukey HSD P22-25 vs Adult: all measures $p < 0.001$).

Animals' median speed, and total distance travelled per trial, changed across developmental time (Supplementary Fig. 2A). Sub-sampling data to match either speed or distance travelled did not affect the developmental trends reported above (Supplementary Fig. 2B–E), ruling out sparser positional sampling early in development as a possible experimental confound. Likewise, although the mean firing rate of Subiculum neurons, and BVCs in particular was notably lower in developing animals (Supplementary Fig. 3A), this could not explain the imprecise spatial firing observed in developing neurons, as sub-sampling spike data to match median firing rates did not affect the developmental trends reported above (Supplementary Fig. 3B, C). Finally, we tested whether poorer spatial firing in development could be due to differences in tetrode recording stability across sampled ages: spike cluster drift was greater in young animals (Supplementary Fig. 4A), but sub-sampling neurons to match cluster drift did not affect reported developmental trends (Supplementary Fig. 4B, C). Spike cluster isolation quality did not change across development (Supplementary Fig. 4A). In summary, although subiculum BVCs are present in pre-weanling animals, adult-like stability and spatial specificity emerge late in development (>P25).

### BVC receptive field spatial tunings in a square environment

We examined the distributions of spatial tuning parameters ($d$, $\sigma_0$ and $\Phi$) for BVCs recorded at each age. At all ages, distance tunings ($d$) were highly skewed towards short distances, although tunings up to half of the arena width were observed (Fig. 3A). The median $d$ of recorded BVCs did not change significantly with age (Fig. 3B; Kruskal-Wallis test, Age: $\chi^2_{(3,364)} = 3.69$, $p = 0.30$). The tuning field widths of BVCs ($\sigma_0$) were distributed over all four tested levels, at all ages (Fig. 3C), and the median $\sigma_0$ did not change significantly with age (Fig. 3D; Kruskal-Wallis test Age: $\chi^2_{(3,364)} = 3.91$, $p = 0.27$). The distribution of directional tunings ($\Phi$) exhibited a striking departure from a uniform distribution: $\Phi$ values were strongly clustered at the cardinal points of the compass, aligned to the orientations of the arena walls, in all age groups (Fig. 3E).

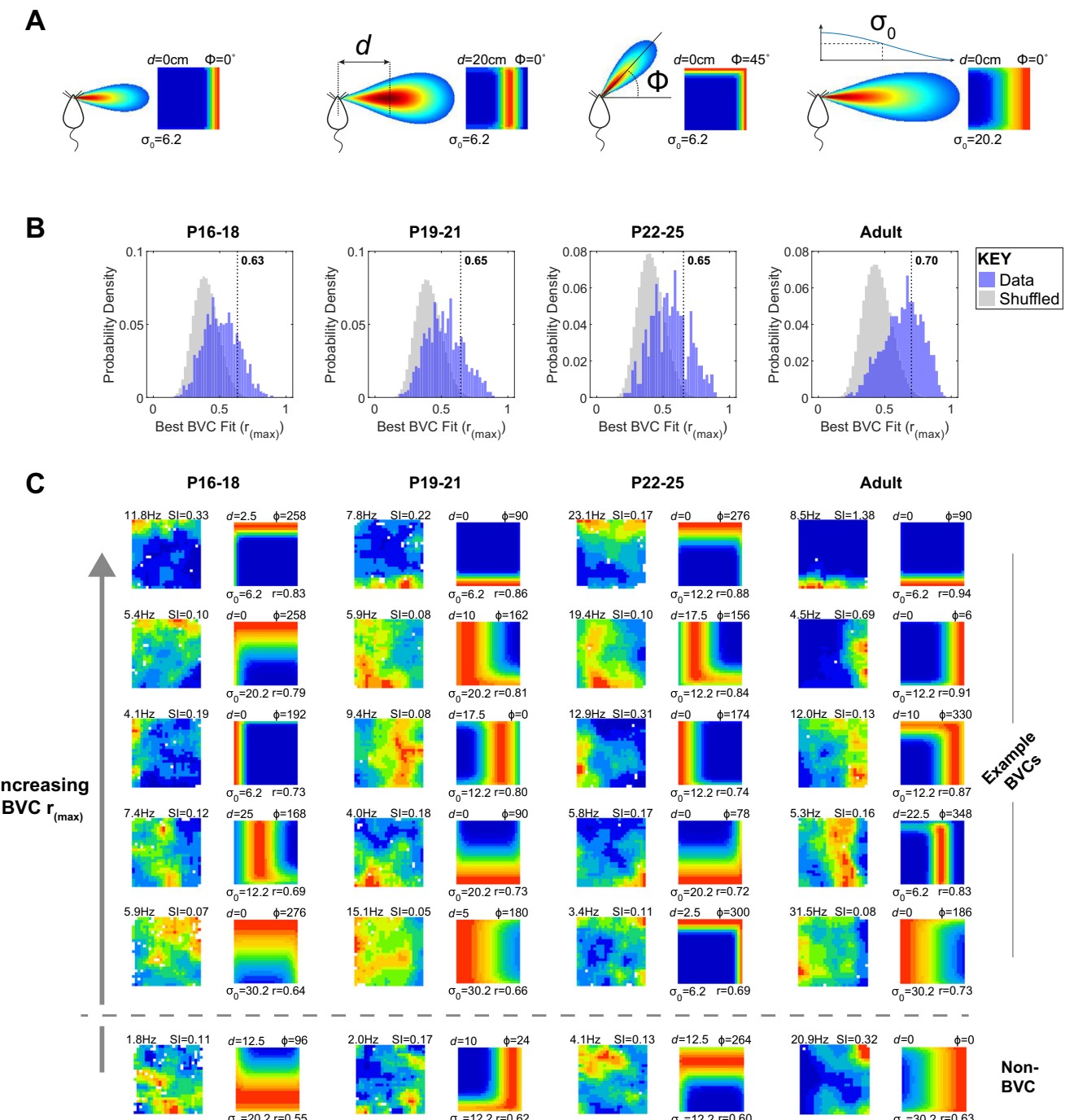

**Fig. 1 | Classification of subiculum neurons as BVCs. A** Example idealised BVC receptive fields and corresponding expected firing rate maps. Left: receptive field cartoon showing positions (relative to the animal) for which the presence of a boundary leads to maximal BVC firing (red), falling off to minimal firing (blue). Middle, right: receptive field schematics and corresponding rate maps showing the effect of changing the three parameters varied in the BVC fitting procedure: the distance tuning, $d$; the allocentric directional tuning, $\Phi$; and the width of the distance tuning curve, $\sigma_0$. $\Phi$ is defined with respect to the environmental reference frame: it is independent of the heading direction of the animal. **B** Distributions of correlations between neuronal firing rate maps and the best-fitting BVC map ($r_{(max)}$) at different ages. Blue histograms show subiculum data $r_{(max)}$, grey histograms show $r_{(max)}$ based on shuffled data. Vertical dashed lines show the population threshold $r_{(max)}$ for BVC classification, defined as the 99th percentile of the spike-shifted $r_{(max)}$ distributions, within each age group. Cells were also classified using a threshold generated for each rate map, see Methods for further details. **C** Neuronal firing rate maps and best-fit BVC model maps, for five example BVCs and one example non-BVC from each age group. Each row shows example BVCs, each column of paired maps is comprised of the neuronal firing rate map (left) and the best-fit BVC model map (right). For both maps, hot colours indicate high firing rates. Columns of paired maps show data from different ages (cells differ across age groups). Text adjacent to the neuronal firing rate map shows peak rate (top left, Hz) and Spatial Information ('SI'; top right). Text adjacent to model map shows the $d$ (top left), $\varphi$ (top right), $\sigma_0$ (bottom left) tuning parameters of the BVC, and $r_{(max)}$ (bottom right). For each age group, representative examples are shown from differing levels of $r_{(max)}$, sorted from highest (top) to lowest above classification threshold (bottom). The bottom row shows a non-BVC from each age group, for which $r_{(max)}$ fell slightly below classification threshold.

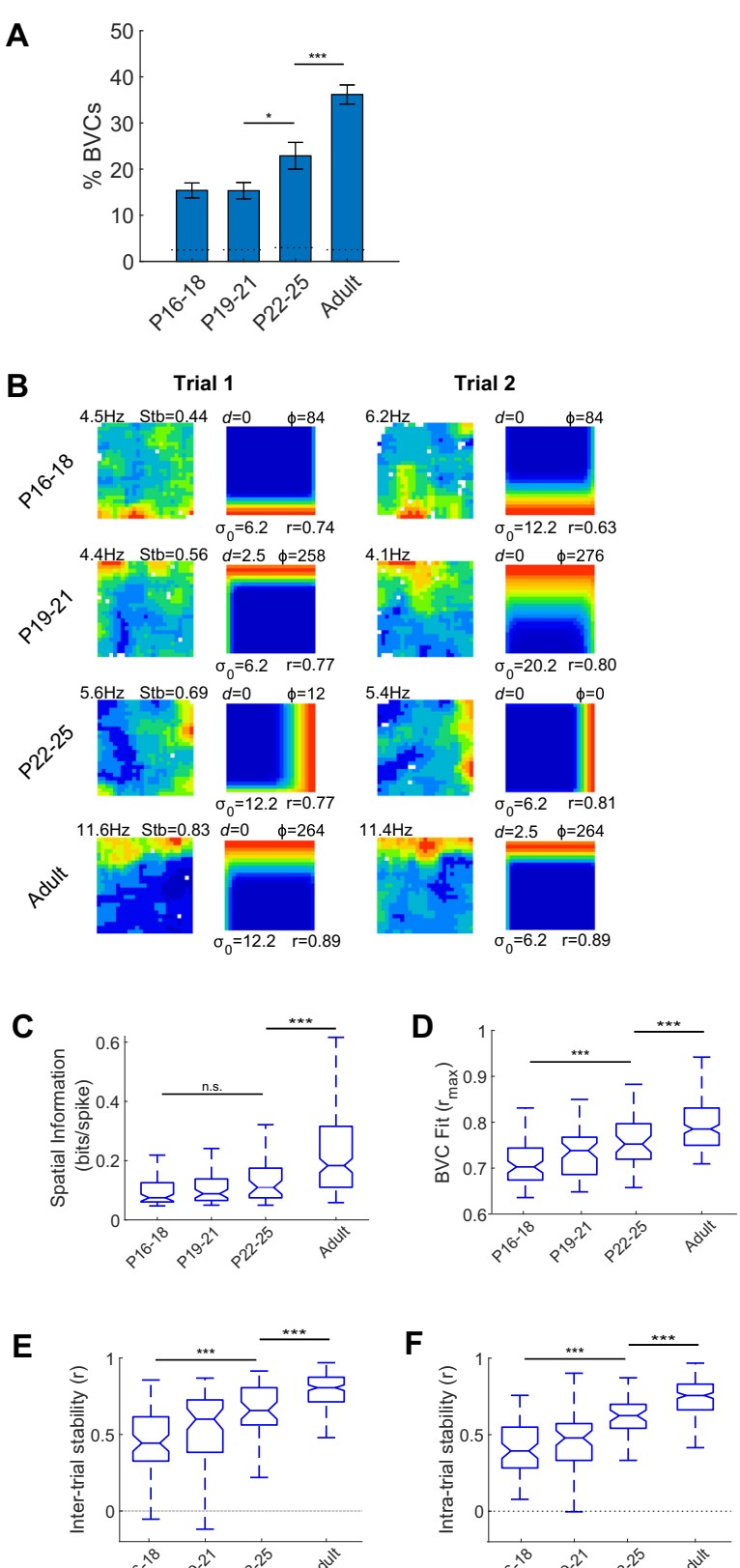

We quantified the four-fold symmetrical clustering using the Rayleigh test on quadrupled, wrapped $\Phi$ values (see methods). At all ages, $\Phi$ showed a significant four-fold departure from uniformity (P16-18, $z = 9.6$, $p < 0.001$; P19-21, $z = 17.0$, $p < 0.001$; P22-25, $z = 11.1$, $p < 0.001$; Adult, $z = 50.7$, $p < 0.001$). To further compare four-fold clustering across ages, we quantified the proportion of BVCs with $\Phi$ oriented ±12°

of a cardinal compass point. As expected, this proportion was well above chance, and did not significantly change during development (Fig. 3F; $\chi^2 = 3.46$, $p = 0.33$).

Four-fold symmetry of BVC $\Phi$ tunings is not related to modulation of BVC firing by allocentric head direction: although BVCs carry slightly more head-directional modulation than non-BVC neurons

**Fig. 2 | Prevalence, spatial specificity, and spatial stability of BVC firing, across development. A** Prevalence of BVCs (percentage of all subiculum neurons classified as BVCs in each age group). Error bars show 95% confidence interval of the proportion. Horizontal dashed lines show 95% confidence level for the number of BVCs exceeding that expected by chance. *** indicates significant differences at $p < 0.001$ level. **B** Example BVCs recorded across two consecutive trials, showing increases in inter-trial stability with age. Each row shows a BVC, each column of paired maps shows the neuronal rate map (left) and best-fit model map (right) for one recording trial. Text adjacent to neuronal firing rate map shows peak firing rate (top-left) and inter-trial stability (top right). Model map format as for Fig. 1C. Inter-trial stabilities for each example lie within SEM of mean, for respective age group. Boxplots showing distributions of Spatial Information (**C**), BVC $r_{(max)}$ (**D**), inter-trial stability (**E**) and intra-trial stability (**F**), for all BVCs in each age group. Boxes show Inter-Quartile Range (IQR), central line shows median, notch shows 95% confidence interval for the median, and whiskers show the extremes of the data, excluding outliers (see Methods for further details). *** indicates difference significant at $p < 0.001$ level.

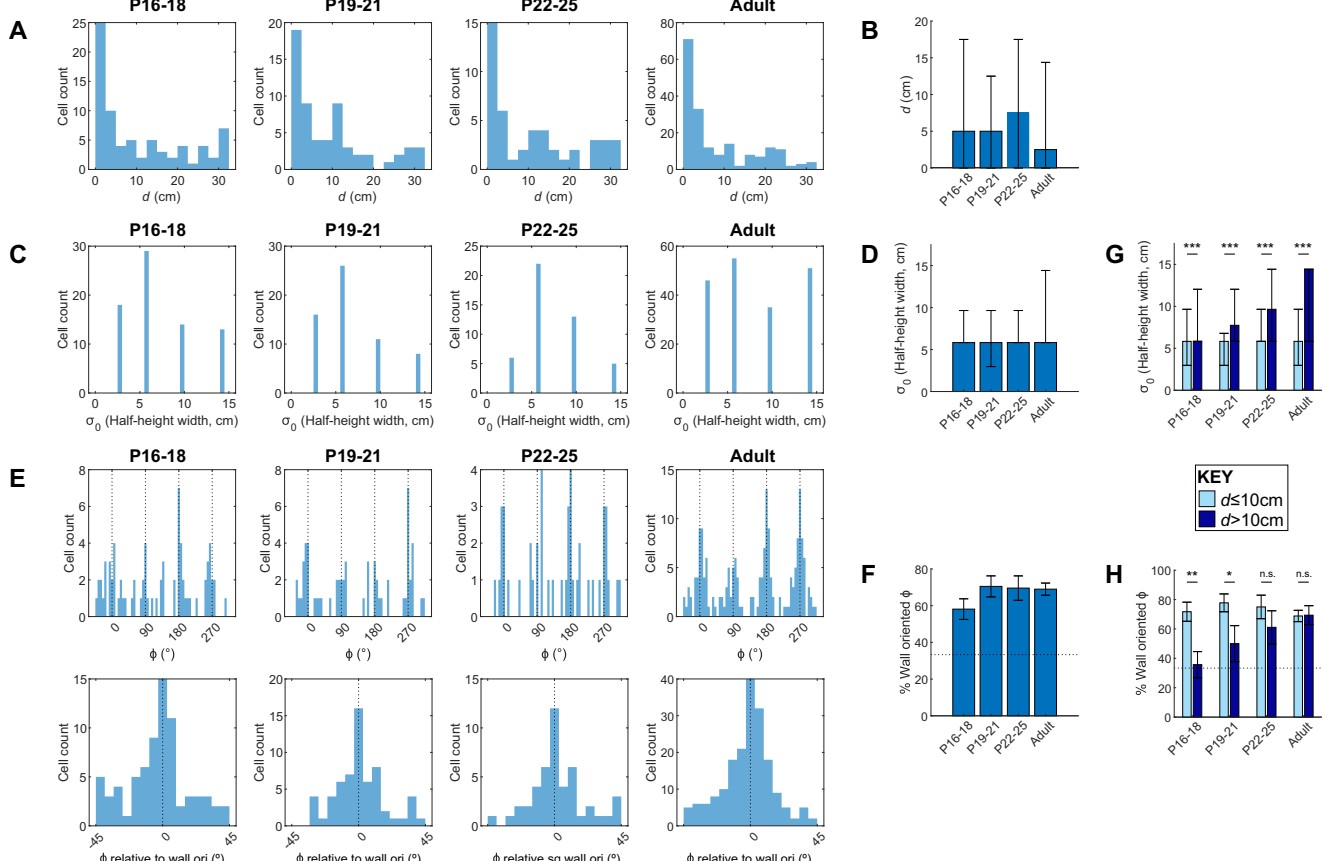

**Fig. 3 | Characterisation of BVC tunings for all subiculum BVCs, in each age group. A** Histograms showing all $d$ tunings in each age group. **B** Median $d$ tunings for each age group. Error bars show 25th and 75th percentiles of $d$. **C** Counts of $\sigma_0$ tunings in each age group. $\sigma_0$ tunings are expressed as the half-height width of a BVC field with given $\sigma_0$, assuming $d = 0$ and $\Phi = 0$. **D** Median $\sigma_0$ tunings for each age group. Error bars show 25th and 75th percentiles of $\sigma_0$. **E** Top row: Histograms showing all $\Phi$ tunings in age group. Black dashed lines show values of $\Phi$ oriented towards arena walls. Bottom row: histograms of all $\Phi$ tunings mapped onto one 90° quadrant. 0° indicates $\Phi$ tunings aligned to a wall. **F** Proportion of BVCs with $\Phi$ oriented towards walls (±12°), in each age group. Error bars show 95% confidence interval. Horizontal dashed line shows expected proportion, assuming a circularly uniform distribution of $\Phi$. **G** Median $\sigma_0$ tunings for each age, split according to short-range (≤10 cm; light blue) and long-range (>10 cm; dark blue) $d$ tunings. Error bars show 25th and 75th percentiles of $\sigma_0$. **H** Proportion of BVCs with $\Phi$ oriented towards walls (±12°), for each age group and split according to short- and long-range $d$ tunings. Error bars show 95% confidence interval for the proportion. *indicates significant differences at $p = 0.05$ level, ** at $p = 0.01$ level, *** at $p = 0.001$ level.

(Supplementary Fig. 5A), the majority of BVC head directional tuning results from position-by-direction sampling bias (Supplementary Fig. 5B). Furthermore, BVC preferred head direction tunings do not display four-fold symmetry (Supplementary Fig. 5C).

To test whether four-fold $\Phi$ symmetry may emerge artefactually from behavioural biases in the square environment more generally, a population of simulated BVCs was produced using real position data and synthetic, mean rate-matched spike trains. In the latter, spike likelihood was based on a combination of both the BVC function evaluated at each location, and a random Poisson process (see Methods). When simulated $\Phi$ tunings were drawn from a flat distribution, no four-fold symmetry in the resulting BVCs was detected (Supplementary Fig. 5D, E). As a positive control, a second simulated BVC

population was generated, but in this case $\Phi$ tunings were drawn from the real data: this population displayed a strong four-fold symmetry, as in real data (Supplementary Fig. 5F, G). These control analyses suggest that the four-fold $\Phi$ tuning symmetry observed in real data is indeed a feature of BVC receptive field tuning, and not an artefact due to position bias in the square arena.

Following this, we tested whether the tuning characteristics of BVCs change depending on their preferred firing distance from a boundary, by splitting BVCs into short- and long-range distance tuning groups ($d \leq 10$ cm and $d > 10$ cm, respectively). Median $\sigma_0$ levels were significantly greater for long-range than short-range BVCs, at all ages (Fig. 3G; Wilcoxon test long vs short, $p < 0.001$ for all age groups). The clustering of $\Phi$ at wall orientations did not differ between long- and

short-range BVCs in adults, but did in developing animals, with $\Phi$ being significantly more clustered at wall orientations for short-range BVCs, up until P21 (Fig. 3H; $\chi^2$ Age*Range: $\chi^2 = 27.4$, $p = 0.002$; $Z$-test for proportions short vs long: P16-18, $p = 0.002$; P19-21, $p = 0.036$; P22-25, $p = 0.32$; Adult, $p = 0.94$). The lack of wall-aligned $\Phi$ tunings in long-range P16-18 BVCs could be due to a sensory deficit: vision is of lower acuity early in development[38], and long-range BVCs are active when animals are further from visual sensory landmarks such as boundaries or external cues, possibly leading to a reduction in the tuning or stability specifically in long-range BVCs. However, there was no age-specific reduction in general measures of spatial firing quality, including Spatial Information, BVC fit $r_{(max)}$ or intra-trial stability in long-range BVCs (Supplementary Fig. 6A–C), arguing against this hypothesis. There was a trend towards lower stability between consecutive trials in P16-18 long-range BVCs (Supplementary Fig. 6D), suggesting memory-based re-instantiation of BVC activity is less stable in young animals in locations far from visual landmarks: however, this would not affect $\Phi$ tuning orientations, within an individual trial. Overall, therefore, the geometry of the population of BVC receptive fields is unchanged between early development and adulthood, with the exception that the striking influence of environment walls on directional tunings does not extend to long-range BVCs, until after weaning.

## BVC receptive field directional tunings change between square and circle

To test whether the clustering of $\Phi$ tunings in the square is caused by the geometry of environment boundaries, we exposed the rats to a circular open arena (diameter 80 cm). The distributions of $\Phi$ and $d$ tunings in both square and circle are illustrated in Fig. 4A, which shows a clear contrast between the distribution of $\Phi$ tunings in the square (clustered at cardinal compass points) and those in the circle, which possess an apparently uniform angular distribution. Indeed, BVC directional tunings in the circular arena showed no significant 4-fold radial symmetry or unimodal departure from uniformity (Fig. 4B; Rayleigh test quadrupled $\Phi$: P16-18, $z = 0.5$, $p = 0.63$; P19-21, $z = 1.5$, $p = 0.23$; P22-25, $z = 0.6$, $p = 0.57$; Adult, $z = 1.8$, $p = 0.16$; Rayleigh test unimodal: P16-18, $z = 2.3$, $p = 0.098$; P19-21, $z = 2.2$, $p = 0.11$; P22-25, $z = 0.4$, $p = 0.65$; Adult, $z = 0.0$, $p = 0.98$). A full description of BVC spatial tuning, stability and receptive field properties in the circle is shown in Supplementary Fig. 7A, B: overall, differences in BVC firing between square and circle are limited to the distribution of $\Phi$ tunings, as described above. The only exception is BVC fit $r_{(max)}$, which is significantly lower in the circle, across all age groups. Crucially, the reduced $r_{(max)}$ in the circle does not lead to changes in $\Phi$ tuning distribution: sub-sampling the circle BVCs to match BVC $r_{(max)}$ across shapes did not affect BVC spatial firing characteristics (Supplementary Fig. 7C, D), or any of the results reported in Fig. 4 (Supplementary Fig. 7E–G).

Inspection of the rate maps reveals that $\Phi$ tunings sometimes rotated between square and circle in the laboratory reference frame, likely due to the lack of common extra-maze cues across these environments (Fig. 4C; see Methods). Within each ensemble (comprising all simultaneously recorded BVCs in one recording session), rotations of individual BVC $\Phi$ tunings were clustered around the ensemble mean rotation for both pups and adults (Fig. 4D; Concentration parameter kappa: P16-25, $k = 2.15$; Adult, $k = 2.23$; $K$-test for differing Kappa [Pup vs Adult]: $f = 1.07$, $p = 0.77$), indicating that BVC ensemble rotations were approximately coherent. However, relative $\Phi$ rotations were not fully rigid across co-recorded cells, with $\Phi$ offsets between cell pairs often changing within the range ±45° (see examples Fig. 4C; median absolute rotation relative to mean: P16-25, 30°; Adult, 22°). This partial plasticity in relative $\Phi$ offsets is consistent with a reorganisation of ensemble receptive fields between circle and square, producing the 4-fold clustering observed in the latter environment.

The rotation of BVC ensembles between square and circle raises an alternative explanation for our results: that 90° $\Phi$ clustering is present in the circle within each simultaneously recorded BVC ensemble, but the orientation of these clusters is inconsistently aligned across experimental sessions, obscuring the clustering when all data are aggregated. To test this possibility, we corrected the $\Phi$ tuning for each BVC by subtracting the mean ensemble rotation (for ensembles with ≥5 co-recorded BVCs). Even following this correction procedure, the distribution of $\Phi$ tunings, in circular environments, showed no significant 4-fold departure from uniformity (Fig. 4E, Rayleigh test quadrupled $\Phi$: P16-25, $z = 0.2$, $p = 0.86$; Adult, $z = 0.2$, $p = 0.80$). Furthermore, even within each co-recorded ensemble (≥5 co-recorded BVCs), BVC $\Phi$ tuning distributions showed a significant reduction of 4-fold symmetry when moving from square to circle, as shown by the length of the Rayleigh vector derived from wrapped, quadrupled $\Phi$ (Fig. 4F; Wilcoxon Test; $p = 0.001$). Features of boundary geometry present specifically in the square, as opposed to the circle, are therefore most likely responsible for the observed 90° clustered distribution of BVC $\Phi$ tunings in square environments.

## BVC response to inserted boundaries

An important characteristic of boundary-driven firing is that the introduction of a new boundary, such as a barrier, into the environment causes a new firing field to emerge[9,23]. To assess the development of barrier-driven firing, a barrier (oriented EW or NS) was inserted into the square arena, and we quantified whether BVC firing increased on the distal side (relative to the existing BVC field), as compared to the proximal side of the barrier (following[30]; see Methods). At all ages, barrier insertion caused an increase in absolute firing rate on the distal side (Fig. 5; percentages of tested BVCs with increased firing rate on distal side: P16-18, 79%; P19-21, 78%; P22-25, 93%%; Adult, 92%), though the difference between distal and proximal firing rates did not reach significance until P19 (Mixed ANOVA: Age*Barr Side $F_{(3,137)} = 10.8$, $p < 0.001$; SME Dist vs Prox: P16-18, $p = 0.068$, P19-21, $p = 0.033$, P22-25, $p = 0.010$, Adult $p < 0.001$). However, when changes in firing rate were normalised to overall firing rates in the barrier and baseline trials, the difference between proximal and distal side firing was significant at all ages (Fig. 5C; Mixed ANOVA: Age*Barr Side $F_{(3,137)} = 21.6$, $p < 0.001$; SME Dist vs Prox: P16-18, $p = 0.012$; all other age groups $p < 0.001$), suggesting that part of the slow development of barrier-driven fields is due to low overall firing rates. For both measures (absolute and normalised firing), distal side firing was significantly less than adult for pre-weaning age groups, but not significantly different to adult in post-weaning animals (SME Dist vs Adult: P16-18, $p < 0.001$ [Abs], $p < 0.001$ [norm]; P19-21, $p = 0.021$ [Abs], $p = 0.031$ [Norm]; P22-25, $p = 0.58$ [Abs], $p = 0.33$ [Norm]).

Notably, in adults, barrier insertion causes a significant inhibition of firing rate on the proximal side of the barrier, relative to baseline level, indicating that barrier insertion causes a reorganisation of the BVC field that goes beyond simply replicating the principal firing field (1-sample $T$-Test versus no change: Absolute rate, $p < 0.001$, Normalised rate, $p < 0.001$). However, this reduction in firing is not observed in any development group ($p \geq 0.57$, all groups). In adults only, proximal side inhibition was stronger in BVCs with broader-tuned receptive fields (Fig. 5A c.f. bottom two rows, 5D-E; 1-way ANOVA Best fit model $\sigma_0$: Absolute Proximal Rate, $F_{(3,79)} = 7.91$, $p < 0.001$; Normalised Proximal Rate, $F_{(3,79)} = 3.41$, $p = 0.022$) and was only consistently significant (across both absolute and normalised rate measures) for cells falling in the two broadest $\sigma_0$ categories (1-sample $T$-Test versus no change: Absolute rate, $\sigma_0 = 6.2$, $p = 0.44$; $\sigma_0 = 12.2$, $p = 0.22$; $\sigma_0 = 20.2$, $p < 0.001$; $\sigma_0 = 30.2$, $p < 0.001$; Normalised rate, $\sigma_0 = 6.2$, $p = 0.24$; $\sigma_0 = 12.2$, $p = 0.001$; $\sigma_0 = 20.2$, $p < 0.001$; $\sigma_0 = 30.2$, $p < 0.001$). Where an adult BVC has a broader baseline field, this is consistently inhibited by barrier insertion, on the proximal barrier side. However, this pattern of results was not observed in developing rats (see Supplementary Fig. 8).

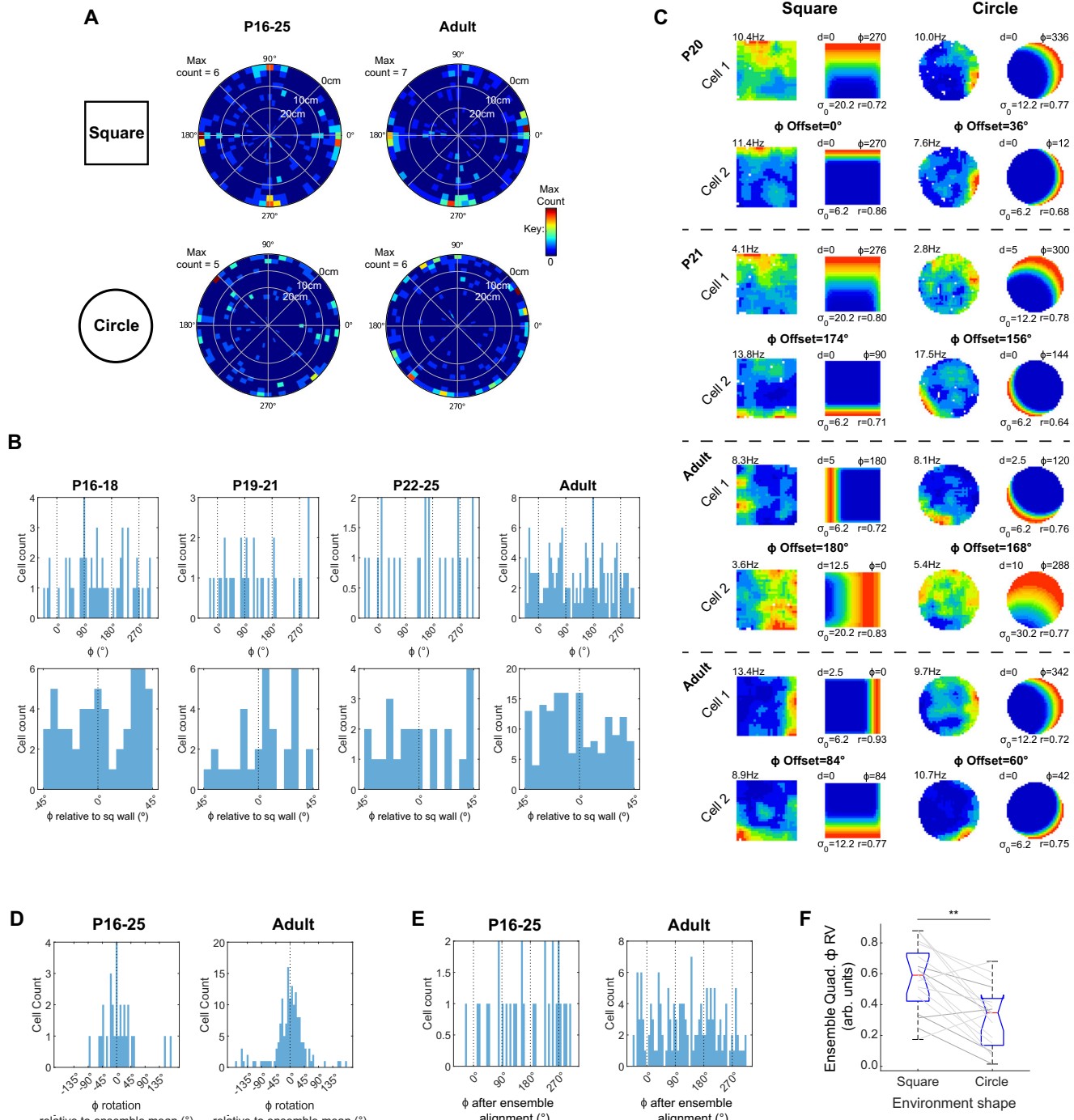

**Fig. 4 | Characterisation of BVC Φ tunings in a circular arena. A** Two-dimensional (polar) histograms showing distributions of BVC $d$ and $Φ$ in the square and circular arenas, in adults and developing rats. Bin colours show histogram counts (hot colours indicate high counts, cold colours low counts), angular axis shows $Φ$, radial axis shows $d$. Radial axis is reversed such that low $d$ is outermost, to enhance visual clarity of clustering of $Φ$ tunings in the square arena. **B** Top row: Histograms showing all $Φ$ tunings in age group, in the circular arena. Black dashed lines show values of $Φ$ orientated towards the square arena walls. Bottom row: histograms of all $Φ$ tunings mapped onto one 90° quadrant. 0° indicates $Φ$ tunings aligned to the square environment walls, in the laboratory reference frame. **C** Neuronal firing rate maps and respective best-fit BVC model maps for four pairs of simultaneously recorded BVCs, in the square and circular arenas. Each row shows

one BVC, dashed lines delineate simultaneously recorded pairs. Text top left of neuronal map shows peak firing rate. Model map format as for Fig. 1C. Text between rows shows offsets of $Φ$ tuning between co-recorded pairs in square and circle. **D** Histograms showing rotations of BVC $Φ$ tunings, relative to the mean $Φ$ rotation for their ensemble (ensembles of ≥5 BVCs only). Left panel shows all developing rats, right panel shows adults. **E** Histograms showing distribution of $Φ$ tunings after alignment to common directional reference frame, by subtraction of ensemble mean $Φ$ rotation from individual BVC $Φ$ rotations (ensembles of ≥5 BVCs only). **F** Rayleigh Vector lengths derived from quadrupled wrapped $Φ$ (quad-$Φ$ RV), for each ensemble with ≥5 BVCs. Boxplots show distribution of quad-$Φ$ RV in square and circle arenas, grey lines show change in quad-$Φ$ RV for each ensemble, between square and circle (light-grey, adult; dark grey, P16-25).

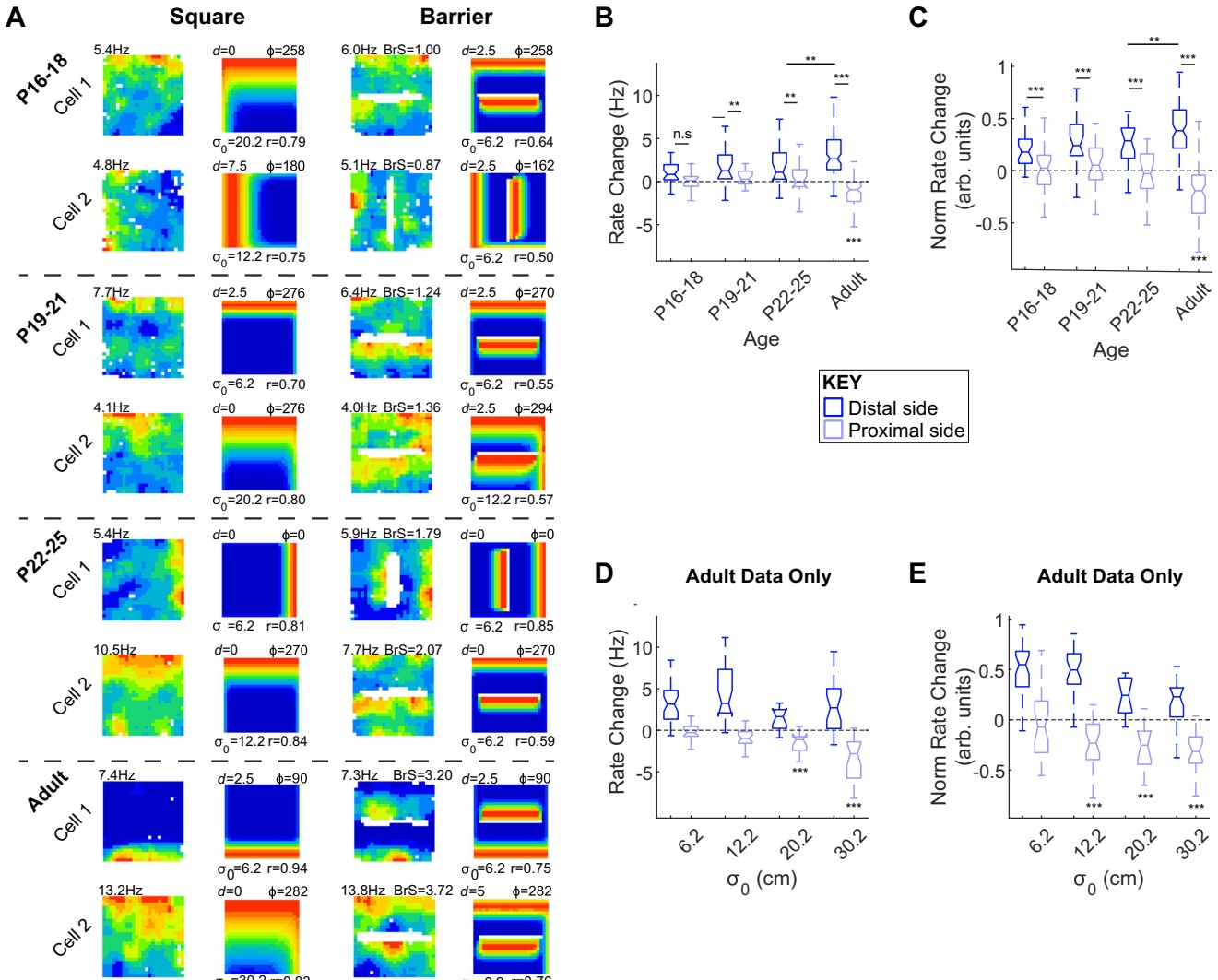

**Fig. 5 | Development of BVC response to inserted boundaries. A** Neuronal firing rate maps and best-fit BVC model maps for eight example BVCs, in the square arena (left columns) and following the insertion of a barrier (right columns). Each row shows one BVC, dashed lines delineate cells recorded at different ages. Text adjacent to neuronal firing rate map shows peak rate (top-left, Hz) or the barrier rate change score (top-right, 'BrS'). Model map format as for Fig. 1C. **B** Boxplots showing distributions of barrier rate change score for BVCs in each age group. Dark blue boxes show rate changes on distal side of the barrier, light blue boxes on the proximal side of the barrier. Boxes show IQR, central line shows median, notch shows 95% confidence interval for the median, and whiskers show the extremes of the data, excluding outliers (see Methods for further details). **C** Boxplots showing distributions of barrier rate change score for BVCs in each age group, normalised to overall rate across square and barrier trials. **D** Boxplots showing distributions of barrier rate change score for adult BVCs, separated by $\sigma_0$ of best-fit BVC in baseline trial. **E** Boxplots showing distributions of normalised barrier rate change score for adult BVCs, separated by $\sigma_0$ of best-fit BVC in baseline trial. *indicates significant differences at $p = 0.05$ level, ** at $p = 0.01$ level, *** at $p = 0.001$ level. Asterisks directly under light blue boxes indicate significance of difference from rate change = 0.

In summary, while excitatory responses to inserted barriers are observed in subiculum BVCs from P16, the inhibitory component of the BVC response observed in fully mature BVCs does not emerge until much later in development.

## Results are robust to changes in BVC classification method

We investigated whether the results reported above are robust to using alternative shuffling methods for BVC definition, for example shuffles that preserve the spatial structure of firing ('field shuffle')[39]. We found that shuffling spatial firing fields, as opposed to spike times, produced higher threshold values of $r_{(max)}$, and a correspondingly smaller population of BVCs, with a more stringent fit to the model. Key study results were unchanged (Supplementary Fig. 9). We also tested the specificity of the BVC model-fitting method by applying it to CA1 data, where BVCs have not previously been reported. BVCs were detected in the CA1 (though less when using the field shuffle than the spike-time shuffle, and always significantly fewer than in Subiculum, in adults; Supplementary Fig. 10A). The majority of BVCs detected from CA1 data took the form of elongated place fields near to environment boundaries (see examples Supplementary Fig. 10B). We found that these spatial responses were better fit by model Place Cells (radially symmetric 2-dimensional Gaussian fields of varying widths, see Methods) than BVCs. On the basis of these results, we introduced a further refinement to our detection procedure, by excluding BVCs whose firing is better fit by a model Place Cell than a model BVC: doing so resulted in a notable reduction in CA1 BVCs, post-weaning and in adulthood (for both spike-time and field shuffles; Supplementary Fig. 10C), and did not change the key results of this study (for spike-time shuffle, Supplementary Fig. 10D–K; for field shuffle Supplementary Fig. 10L–S). Similar proportions of residual BVCs in CA1 and

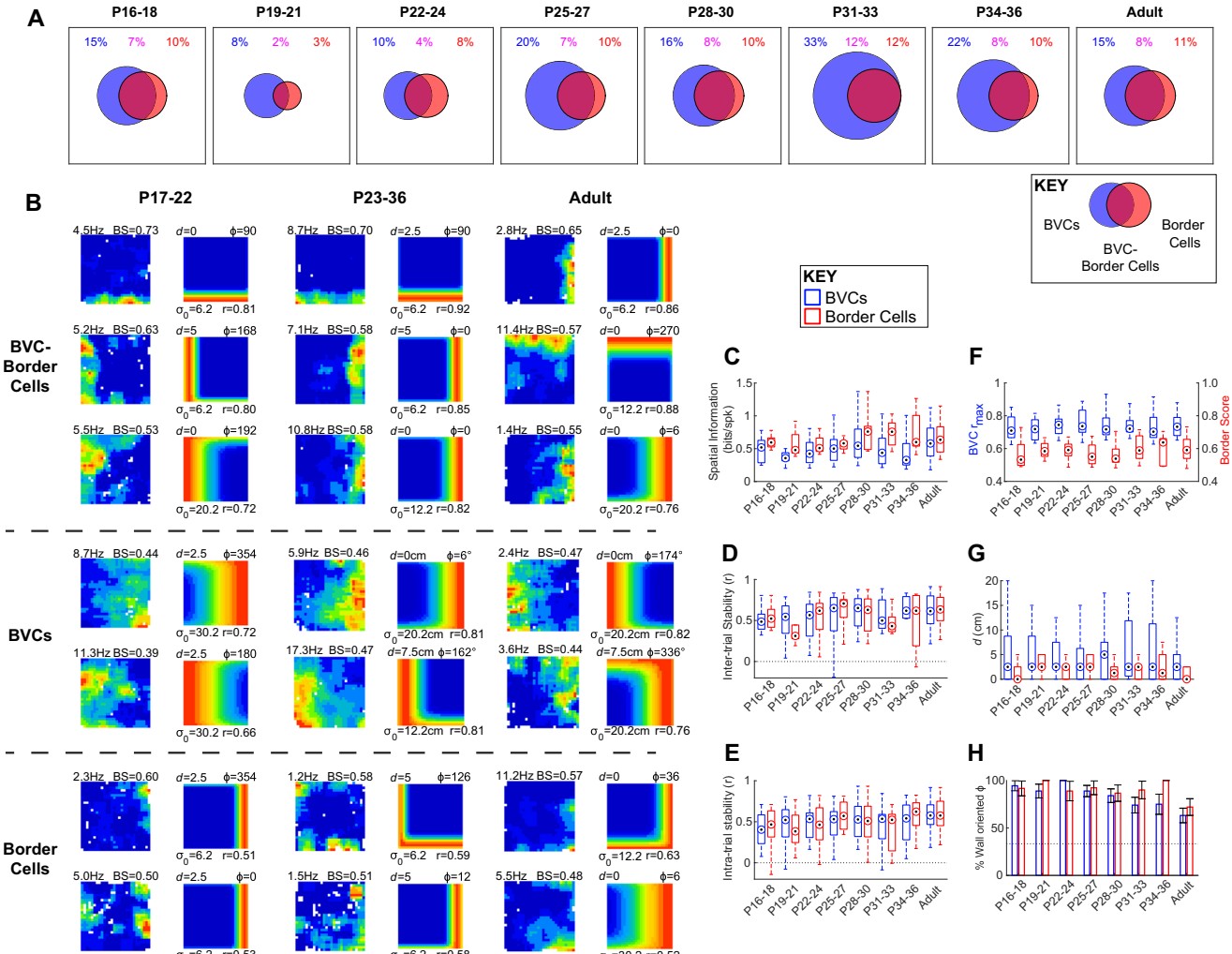

**Fig. 6 | Comparison of BVCs and border cells recorded in medial entorhinal cortex. A** Venn diagrams showing the proportion of mEC cells classified as BVCs (blue), Border Cells (red), or both cell types (overlap region), for each age group. Coloured text above circles shows corresponding numerical percentages. Circles are scaled such that the square bounding box represents 100% of cells recorded in an age group. **B** Neuronal firing rate maps and best-fitting BVC model maps, for example BVCs and Border Cells. Each row shows example cell, columns of paired maps show data from different age groups (cells differ across age groups). Text adjacent to the neuronal firing rate map shows peak rate (top left, Hz) and Border Score ('BS'; top right). Model map format as for Fig. 1C. Top three rows show cells classified as both BVCs and Border Cells, middle two rows cells classified as BVCs but not Border Cells, bottom two rows cells classified as Border Cells but not BVCs.

Boxplots showing distributions of Spatial Information (**C**), inter-trial stability (**D**) and intra-trial stability (**E**), for BVCs (blue boxes) and Border cells (red boxes) in each age group. Cells classified as both BVCs and Border Cells are included in both groups. Box shows IQR, circular target shows median, whiskers show limits of data excluding outliers. **F** Boxplots showing distributions of the BVC $r_{(max)}$ of BVCs (blue boxes; left y-axis) and Border Score of Border Cells (red boxes, right y-axis). Boxplot format as for (**C**). **G** Boxplots showing distributions of $d$ tunings for BVCs (blue bars) and Border Cells (red bars), in each age group. Boxplot format as for (**C**). **H** Proportion of BVCs (blue bars) and Border Cells (red bars) with $\Phi$ oriented towards walls (±12°), in each age group. Error bars show 95% confidence interval for the proportion. Horizontal dashed line shows proportion expected, assuming a circularly uniform distribution of $\Phi$.

Subiculum can be observed in pre-weanling animals across all classification methods; this likely reflects the previously reported dependence of stable CA1 place cell firing on boundaries, in pre-weanling rats[29]. Overall, introducing more stringent criteria for BVC selection did not alter any of the key findings reported here, demonstrating their robustness.

### BVCs and border cells in medial entorhinal cortex are mature at P17

We have shown that subiculum BVCs, although present at P16, continue to mature late into development (>P25), with spatial tuning, stability, and barrier responses all remaining immature as late as P25. This is in contrast to previous reports of mEC border cells[30] which showed no change in either spatial tuning or stability between P17 and adulthood. These contrasting results could be due to the faster

maturation of mEC (as suggested by[37]). However, an alternative possibility is that the boundary-responsive cells captured by the BVC measure mature more slowly than those defined using the border score (which is biased towards selecting fields near to boundaries). To rule out this latter possibility, we re-analysed the data described by ref. 30, and quantified the development of boundary-driven firing using either the border score, or the BVC classification method (as described here) to define boundary-responsive neurons.

Figure 6A shows the proportion of mEC neurons classified as either a border cell, a BVC or both. Distributions of shuffled and actual BVC fits to the data are shown in Supplementary Fig. 11A. As previously reported, there is no difference between the proportions of mEC border cells in adults and developing rats (Z-test proportions adult vs developing: Z = −1.02, p = 0.31), and the same is true for the proportion of BVCs (Z = 0.26, p = 0.80). At all ages, there is considerable overlap

between those boundary-responsive neurons classified as border cells and BVCs, but the proportion of these "intersect" cells does not change between adults and developing rats ($Z = -0.88$, $p = 0.38$). Figure 6B shows examples of neurons which were classified as BVCs, Border Cells, or as both types simultaneously. Cells classified as both BVCs and border cells show clear firing fields close to, and extending along, boundaries. Cells selected uniquely by the BVC measure show broader firing fields with firing extending away from walls. Neurons classified as border cells, but not BVCs, by contrast, exhibit multiple firing fields which are close to walls, but do not cover their full extent. For all cell types, the qualitatively-judged firing properties do not change with age.

We confirmed the precocious maturity of mEC boundary-related firing by quantifying the spatial information and stability of the classified cells. The spatial tuning, stability and BVC model fit of mEC BVCs does not change during development (Fig. 6C–F, ANOVA Age: Spatial Information, $F_{(7,156)} = 1.8$, $p = 0.09$; Inter-trial stability, $F_{(7,151)} = 0.84$, $p = 0.56$; Intra-trial stability, $F_{(7,156)} = 1.28$, $p = 0.26$; BVC $r_{(max)}$, $F_{(7,156)} = 0.48$, $p = 0.84$), and furthermore, we confirmed that neither does the tuning, stability and Border Score of border cells (Spatial Information, $F_{(7,84)} = 0.05$, $p = 0.40$; Inter-trial stability, $F_{(7,85)} = 1.8$, $p = 0.10$; Intra-trial stability, $F_{(7,84)} = 0.99$, $p = 0.44$; Border Score, $F_{(7,84)} = 0.72$, $p = 0.66$). Boundary-responsive firing in the mEC appears, therefore, mature from P17, irrespective of the measure used to quantify it. Similarly, the tuning properties of mEC BVCs do not change throughout development: though BVCs have longer distance tunings than border cells (Fig. 6G, Supplementary Fig. 11B, E; Wilcoxon test BVC vs Border Cell all ages: $p < 0.001$), the median distance tunings of either does not change during development (Kruskal-Wallis Age: BVCs, $\chi^2_{(7,156)} = 5.4$, $p = 0.61$; Border Cells, $\chi^2_{(7,84)} = 11.8$, $p = 0.11$). The proportion of angular tunings oriented to walls is significantly higher than chance, at all ages (Fig. 6H, Supplementary Fig. 11G), and does not change with age ($\chi^2$ Age: BVCs, $\chi^2 = 56$, $p = 0.23$; Border Cells, $\chi^2 = 48$, $p = 0.24$). The distortion of boundary-responsive angular symmetry is also apparent in mEC boundary-responsive neurons, from early in development (though responses to a circular arena were not tested within this dataset).

The results above demonstrate that BVCs and Border Cells in mEC are already mature by P17, in contrast with the extended maturation (>P25) displayed by subiculum BVCs. The spatial tuning and stability of spatially-modulated subiculum neurons which did not fit BVC criteria (Supplementary Fig. 12A–C; see Methods) were also immature at P25. This suggests that the slow development of subiculum circuitry is a more general phenomenon and is not limited to BVC neurons. Furthermore, both mean rate (Supplementary Fig. 3) and burst-firing tendency (Supplementary Fig. 12D, E) of subiculum neurons mature during or after the period P16-25, suggesting that the subiculum, as a whole, displays a protracted maturational trajectory.

## Discussion

This study is the first to define a population of BVCs in the subiculum and characterise their spatial tuning. Previous reports of BVCs in the subiculum did not use a formal objective procedure to define a BVC[10,26], and the method used to define Border Cells in mEC[23,30] selectively weights firing fields in close proximity to boundaries and is not well-suited to capture the broader and vectorial nature of spatial firing in the subiculum. The BVC model fit method used here captures both long- and short-range boundary tunings, and our parallel analysis of mEC and subiculum data confirms that mEC boundary-driven cells respond selectively closer to walls, whereas in the subiculum a broader range of distance tunings are present (cf Figs. 3B, 6F). Subiculum neurons whose spatial tuning closely fits that of idealised BVCs have previously been reported[33], using a similar method to our own, though this study did not provide any further characterisation of the BVCs detected. The goodness-of-fit r-values appeared markedly lower in that

study than those reported here: this may be explained by the fact that[33] used a wall-less arena, which can reduce the spatial specificity of BVCs[9]. We introduce two modifications to our method, both of which restrict the population of BVCs to those with a more stringent match to canonical BVCs: using the field shuffle to generate $r_{(max)}$ thresholds and excluding those BVCs whose firing is better fit by a model place cell. For the purposes of this study, which required the characterisation of immature BVCs, a more inclusive selection method was chosen. However, here we have also characterised the analytical tools (the two more stringent selection methods outlined above) best suited to studies focused on adult BVC responses.

The model-fitting method used here allowed us to characterise the population of detected BVCs in terms of the preferred distance-to-boundary ($d$), allocentric angle-to-boundary ($\Phi$) and receptive field width ($\sigma_0$) of the idealised BVCs fitted to neural data. Here we show that preferred distance tunings are biased towards short-range tunings (modal $d$ was 0 cm, at all ages), but the full range of possible preferred distances were detected, up to the arena radius. The preferred distance-to-boundary tunings described here are smaller than recently reported object-responsive subiculum neurons[34] (Vector Trace Cells $14.1 \pm 1.0$ cm [Mean ± SEM], Non-trace cells $11.2 \pm 0.4$ cm; Adult BVCs in present study, $7.5 \pm 0.6$ cm), which may reflect the smaller arena used in this study (62.5 cm, versus 100 cm) preventing the detection of longer-range BVCs. Nonetheless, this study confirms the vectorial nature of boundary coding in the subiculum and builds on previous reports of object vector coding[34,40–42] to suggest that the vectorial coding of allocentric space is fundamental to hippocampal spatial cognition.

A striking feature of BVC tunings is the non-uniformity of the receptive field preferred angles in a square arena, which were strongly clustered with a four-fold circular symmetry, aligned with walls of the square environment (as predicted by ref. 43), whereas this four-fold clustering is not present in a circular environment. BVC receptive fields are unlikely, therefore, to solely represent an allocentric vector to boundaries[25], but are instead further modulated by the geometric features of those boundaries (e.g. straight walls and corners). This result is consistent with both the well-established finding that the geometry of walls and corners influences spatial memory[17–19,22], and that corners in particular may be an important class of geometric cue, with acute corners being more salient than obtuse corners[44,45]. The modulation of BVC firing by boundary geometry, and corners in particular, may therefore represent a mechanism by which these geometric features exert effects on behaviour. Future research could focus on the question of how straight walls and corners alter BVC receptive fields. The existence of a sub-set of subiculum neurons that specifically encode convex and concave corners[46] offers a potential neural mechanism underlying BVC firing modulation by geometric cues; however, the interactions between these cells and BVCs remain to be understood.

The distortion of BVC firing by environment geometry echoes previous findings of distortion of grid cell firing patterns[47,48]. Our study confirms that boundary firing in both subiculum and mEC aligns with individual walls, supporting the hypothesis[47] that grid field shearing may be caused by a combination of grid shrinkage[49] and anchoring of the grid to a sub-set of local environment cues (e.g., one boundary of the environment). A further example of environmental geometry influencing spatially modulated firing is the fact that square environments enforce a four-fold symmetry in the directional tuning curves of egocentric boundary cells in the retrosplenial cortex[50], which the authors speculate may create boundary-driven distortions in downstream allocentrically-tuned neurons.

Are the effects of boundary geometry on BVCs mediated by other spatial responses? BVCs emerge in development earlier than grid cells[30,51,52] suggesting that the influence of geometry on BVC $\Phi$ tunings is independent of grid cell firing and is not simply a downstream

reflection of shearing of grid cells in square arenas. The ontogeny of egocentric boundary cells (ECBs) is currently unknown, so it is not possible to draw conclusions regarding interactions between these cells and BVCs on the basis of their relative timelines of emergence during development. We note, however, that the receptive fields of EBCs are *not* altered by geometry: four-fold directional symmetry emerges as a direct consequence of square geometry constraining the head orientations for which rats can occupy a specific egocentric bearing to the boundary (see Fig. 5 in ref. [50]). In our study, instead, we have ruled out any possible effects of behavioural biases along square walls and shown that the spatial receptive field of a BVC itself is altered by environment geometry. We therefore conclude that distortion of BVC firing in square environments is a conceptually distinct phenomenon from the four-way head direction tuning of EBCs.

Boundary geometry can affect spatial memory in ways that recapitulate its effects on spatially-modulated neurons: distance estimates are distorted in a rhomboid environment, particularly at the narrow end[53]; shrinking or stretching a square environment shifts goal search locations in a way predicted by place field movements[22]. If BVCs encode an allocentric map of position with respect to current boundaries (as in [54] for example) then distortion of BVC receptive fields by geometry may lead to inaccurate navigation following a switch between environments of different shapes. Consistently with this hypothesis, moving between a square- to circular-bounded watermaze (whilst keeping all extra-maze cues constant) disrupts memory for platform position[55], whilst shape transfers that preserve local geometry of corners and walls in specific parts of the maze also preserves memory of these locations, in a hippocampally-dependent fashion[56,57]. Given the nature of the BVC receptive field distortion we report here, we can make a specific prediction related to the hypothesis above: the behavioural effects of switching shape will be maximal at environmental corners (where the BVC representation changes substantially between circle and square) and minimal/negligible in locations near the middle of walls (where the BVC representation changes relatively little when switching shapes).

The inhibition of adult BVC firing on the proximal side of an inserted barrier is a further property of BVCs not predicted by the canonical model[25], which predicted the existence of solely excitatory receptive fields. Inhibitory receptive fields have also previously been seen in unpublished observations (PhD thesis; Shailendra Rathore, Neil Burgess, Francesca Cacucci). Proximal side inhibition of BVC firing offers a possible explanation for the observed coherence of the BVC representation of space: BVCs do not remap between different environments[26], and if their directional tunings rotate, they do so in unison[58] (see also this study). A hypothesis put forward to explain such coherence is that BVCs form a continuous attractor representation of allocentric position, analogous to the directional continuous attractor formed by head direction and grid cells[59–63]. Proximal side inhibition may reflect BVCs (directly or indirectly) inhibiting other BVCs with directionally-opposed receptive fields, thus reflecting connectivity patterns which are necessary for attractor dynamics to emerge[59].

In addition to describing the adult subiculum BVC population, here we also offer the first characterisation of the emergence of subiculum BVCs during post-natal development, and specifically across weaning, when hippocampal learning and memory first emerges[64]. We found that the development of BVCs has both precocious and late-emerging aspects: the spatial stability and specificity of BVCs, and their responses to inserted barriers, develops gradually and slowly, mirroring the previously described gradual emergence of place cell firing in CA1[51,52,65]. This pattern of development provides a notable contrast with boundary-driven firing in the mEC, which appears adult-like from P17 onwards[30]—a finding we have confirmed in our reanalysis of mEC data. These data are consistent with the proposal that maturation of different hippocampal regions proceeds sequentially, along the tri-synaptic loop, with entorhinal areas maturing first, followed by CA3,

CA1 and subiculum[37]. The slow maturation of BVC specificity is consistent with the late-emerging ability of boundaries to enable accurate recall of spatial locations[66,67], and suggests that BVC development may be a key limiting step in the late emergence of place learning more generally[68,69]. Stable spatial signalling close to boundaries emerges early in the mEC (Border Cells[30] and short-range BVCs, see Fig. 6). One prediction of our findings, therefore, is that the development of accurate navigation should emerge earlier for goals close to, as opposed to far from, boundary locations.

In other respects, however, the development of BVCs is precocious. BVC firing is present at the earliest ages tested, and the overall characteristics of BVC spatial tunings, at the population level, appear in adult-like form: average $d$ and $\sigma_0$ do not change with age, and $\Phi$ tunings are clustered to align with wall orientations in the square. The core geometrical properties of the BVC representation of space thus appears in an already mature form at the earliest age tested (P17). The only exception to this pattern is the distorting influence of boundaries on BVC directional tunings for BVCs with long distance tunings, which only emerges post-weaning, a finding which echoes increases in place cell stability far from walls at the same age[29]. The presence of a specific geometry of BVC tunings for square environments may be a neural mechanism enabling the recognition of geometric boundary information, thereby underlying the early emergence of spatial reorientation[18,70,71]. It is unclear whether this mechanism is innate or experience dependent. We note that the animals in this study were reared in rectangular cages: it remains to be established how BVCs would respond following rearing in circular environments, a manipulation which impairs the use of geometrical information in rodents[70].

## Methods

### Subjects

Subiculum data was collected from 17 developing male outbred (lister hooded) rat pups (aged P12-P14 and weighing 24–32 g at time of surgery) and 4 male (3-6mo) lister hooded rats. Developing rats were bred in-house and remained with their dams until weaning (P21). Adult experimental animals and breeding for litters were obtained from Charles River (UK). Male rats, only, were used such that the oestrous cycle stage of female rats was not an uncontrolled variable in experiments. Rats were maintained on a 12:12 h light:dark schedule (with lights off at 10:00). At P4, litters were culled to 8 pups per mother in order to minimise inter-litter variability. Pregnant females were checked at 17:00 daily and if a litter was present, that day was labelled P0. After surgery (see below), each pup was separated from the mother for between 30 min and 2 h each day, to allow for electrophysiological recordings. All experiments were conducted in compliance with UK legislation ASPA (1986), and were approved by the UCL ethical review panel. The PPL numbers were 70/8636 and 70/7136.

### Surgery and electrode implantation

Rats were anaesthetised using 1–2% isoflurane, and 0.15 mg/Kg bodyweight buprenorphine. Rats were chronically implanted with microdrives loaded with 8 tetrodes (HM-L coated 90% platinum-10% iridium 17 μm diameter wire). Microelectrodes were aimed at the hippocampal subiculum region. In developing rats, the co-ordinates used were 4.4 mm posterior and 1.3 mm lateral to Bregma, 2.7 mm below brain surface. For adult rats, the co-ordinates used were 5.4 mm posterior and 1.5 mm lateral to Bregma, 2.7 mm below brain surface. After surgery, rats were placed in a heated chamber until they had fully recovered from the anaesthetic (10–30 min) and were then returned to the mother and littermates.

### Single-unit recording

Rats were allowed a 1-day postoperative recovery (1 week for adults), after which microelectrodes were advanced ventrally by 62–250 μm/day until they reached the subiculum cell layer, identified on the basis

of a prominent theta (5–8 Hz) LFP rhythm and the presence of theta-modulated pyramidal cell firing, at which point recording sessions began. Single units in the subiculum were defined as excitatory pyramidal cells on the basis of a waveform width >300 μs. Single-unit data was acquired using an Axona (Herts, UK) DACQ system. LEDs were used to track the position and directional heading of the animal. Isolation of single units from multi-unit data was performed manually on the basis of peak-to-trough amplitude, using the software package 'TINT' (Axona, Herts, UK). Rat position was recorded by tracking two LEDs attached to the headstage amplifier. Electrodes were moved at least 50 μm (normally 100 μm) every day. Cells recorded on different days were treated as independent samples.

## Behavioural testing and environments

Single-unit activity was recorded while rats searched for drops of soya-based infant formula milk randomly scattered on the floor of two different open field arenas: (1) a square-walled (62.5 cm sides, 50 cm high) light-grey wooden box, placed on a black Perspex floor, (2) a circular-walled (80 cm diameter, 50 cm high) light-grey wooden box, placed on a black painted wooden floor. From the square arena, distal visual cues were available in the form of the fixed apparatus of the laboratory. The circular arena was placed within a set of closed black curtains, within which there was only one prominent distal visual cue, an A0-sized white card, illuminated by a 20W (incandescent filament) desk lamp. Before recording in the circular arena, rats were carried through the curtains in a closed black box, which was moved directly between the arenas, without being rotated. Two consecutive square arena trials were always run at the beginning of every experimental session. In addition, rats were familiarised to the square arena for two trials each over 2 days, before recordings began. One circle arena trial (only) was run in an experimental session, either directly after the square trials, or after the barrier trials (see below). To test responses of subiculum neurons to barriers, a single straight barrier was inserted into the standard square arena, aligned parallel to the arena walls, centred within the open field in both x and y dimensions. In the majority of recording sessions (83% of recorded BVCs), both possible barrier orientations (North-South and East-West) were tested. Otherwise, only one orientation was tested, based on the orientation of BVCs as assessed in the baseline (open field square) trials. Barriers were made of the same material and of the same appearance as the arena walls. The majority of rats were tested using a barrier of dimensions 50 cm length, 2.5 cm width and 50 cm height; 4 developing rats were instead tested using a barrier of 40 cm length, 5 cm width and 50 cm height. Trials were 15 min long. Between each trial, the rat rested for a 15-min inter-trial interval in a 25 cm × 25 cm holding box containing a heated pad.

## Construction of firing rate maps

All spike and positional data were filtered so as to remove periods of immobility (defined as speed <2.5 cm/s). Following this, positional data were re-scaled to a standardised size and shape, for consistent comparison to BVC model maps. For the square arena, edges of the arena along each wall were defined as the line of camera pixels (each pixel being 2.5 mm wide) furthest from the centre of the environment where the summed dwell time was ≥1 s. Data outside these edges was discarded, and the remaining data was divided into a 25 ×25 grid of evenly spaced spatial bins, each representing an area 2.5 cm × 2.5 cm. For circular arena data, the centre of the arena was first estimated as being the mid-point between the visited edges of the environment at the cardinal points of the compass, which were defined as the line of camera pixels furthest from the centre where the summed dwell time ≥0.2 s. Following this, all positional data was rotated by 45° around the estimated centre, and the edges and centre were defined again, following the method described above, for the rotated data. These steps were continued in an iterative fashion until consecutive estimates of

the centre converged to within 0.75 cm. Following the definition of the centre of the environment, the edge of the environment was defined as the largest pixel-wide circumference around the centre for which the total summed positional dwell time was ≥1 s. Data outside these edges was discarded, and the remaining data was divided into a 32 ×32 grid of evenly spaced spatial bins each representing an area 2.5 cm × 2.5 cm.

Following standardised scaling of the data, total positional dwell time and spike count for the whole trial was calculated for each spatial bin. The binned position dwell time and spike count maps for each cell were then smoothed using a boxcar filter, 5 × 5 bins. Trials during which the rat visited less than 80% of the total number of spatial bins were not used for further analysis.

## Construction and fitting of BVC model maps

Firing rate maps corresponding to the activity of an idealised BVC were defined following[25]. Briefly, the firing rate $g$ of a BVC tuned to respond maximally to the presence of a boundary segment at distance $d$ and allocentric direction $\Phi$ was defined as:

$$g_{d,\Phi}(r,\theta) \propto \frac{\exp\left[\frac{-(r-d)^2}{2\sigma_{rad}^2(d)}\right]}{\sqrt{2\pi\sigma_{rad}^2(d)}} \times \frac{\exp\left[\frac{-(\theta-\Phi)^2}{2\sigma_{ang}^2}\right]}{\sqrt{2\pi\sigma_{ang}^2}} \quad (1)$$

where $r$ is the distance and $\theta$ the allocentric direction from the animal to the boundary segment, $\sigma_{ang}$ is constant, and radial field extent $\sigma_{rad}$ varies linearly with

$$\sigma_{rad}(d) = \left(\frac{d}{\beta+1}\right)\sigma_0 \quad (2)$$

The firing rate $f(x,y)$, at any position $(x,y)$, of a BVC with receptive field $g_{d,\Phi}$ can therefore be defined by summing Eq. 1 over all directions $\theta$ in steps of size $\Delta\theta$:

$$f(x,y) = \sum_{\theta=0,\Delta\theta..}^{2\pi} g_{d,\Phi}(r_b(\theta),\theta)\triangle\theta \quad (3)$$

where $r_b(\theta)$ is the distance to the first boundary segment in direction $\theta$ (if there is no boundary in that direction $r_b$ is infinite and no firing results). Equation (3) was then evaluated at a series of points corresponding to the centres of an evenly spaced 25 × 25 grid of 2.5 cm × 2.5 cm spatial bins to give the firing rate map for the BVC. The boundary of the environment was defined as falling at the edge of the outermost bins of the grid. For the circular arena, the boundary was defined as a 80 cm wide circle, centred within a 32 × 32 grid of 2.5 cm × 2.5 cm bins, and function (3) was only evaluated at bins whose centres lay within the boundary. In both environments, $\Delta\theta$ was no smaller than 5.7°.

The best-fitting BVC for each neural rate map was defined by varying the values of $d$, $\Phi$ and $\sigma_0$ and conducting an exhaustive search that maximised the Pearson's r correlation between the BVC map and the neural rate map. $r_{max}$ was defined as the correlation between the rate map and the best-fitting BVC map. $d$ varied between 0 cm and 32.5 cm (40 cm in circular arenas), in 2.5 cm steps, $\Phi$ varied between 0° and 354°, in 6° steps, and $\sigma_0$ could take the values 6.2 cm, 12.2 cm, 20.2 cm or 30.2 cm. In total the search set consisted of 3120 BVC model maps in square environments (3840 in circular arenas). Following[25], the value of $\sigma_{rad}$ was held constant at 0.2 radians, and $\beta$ at 183 cm.

## Assessing statistical significance of BVC fit to neural data

BVC fits to neural rate maps were defined as significant on the basis of a comparison to Pearson's-r-values derived from fitting model maps to spatially shuffled neural data. Spatially-shuffled neural data was produced by shifting the spike train by a temporal offset, which was

between 20 s, and trial duration minus 20 s. Rate maps were then constructed using the method described above ("Construction of Firing Rate Maps"). Spatial shuffling was repeated 1000 times for each cell on each trial, using 1000 equally spaced temporal offsets. Model BVCs were fit to these spatially shuffled rate maps as described above ("Construction of BVC model maps"), and the maximum Pearson's-r fit value ($r_{(max)}$) was found. To be classified as a BVC, the $r_{(max)}$ value for each cell had to surpass both of: (1) the 99th percentile of the 1000 shuffled $r_{(max)}$ values generated for that specific cell and trial ('rate map threshold') and (2) the 99th percentile of an aggregated population of shuffled $r_{(max)}$ values ('population threshold'), from all cells recorded in the same brain area and age group. An additional threshold, determining a minimum level of spatial specificity for BVCs was also used, which was defined as the 75th percentile of Spatial Information scores (see below, "Assessing spatial tuning and stability of BVCs") derived from the same population of shuffled data. The 75th percentile was used as the intended purpose of the spatial information threshold was not to reject a null hypothesis of non-spatial firing, but instead set a requirement for a BVC to demonstrate minimal spatial selectivity. The threshold Spatial Information values used were: P16-18, 0.0467; P19-P21, 0.0487; P22-25, 0.0489; Adult, 0.0580. Spatially-modulated neurons which were not BVCs (see Supplementary Fig. 10) were defined as those whose Spatial Information score surpassed the above described threshold Spatial Information, but whose BVC $r_{(max)}$ score was less than both rate map and population BVC $r_{(max)}$ scores.

Neurons were classified as BVCs if they satisfied the above criteria on either of two trials run in the square open arena. The false positive classification rate for each neuron was assumed to be 1.49%, calculated under the assumptions that the chances of satisfying the BVC fit and Spatial Information criteria were independent, and that the chances of satisfying the criteria in either of the two square open field trials were independent. The 95% significance level for the percentage of neurons classified as BVCs at any given age was calculated as the 95th percentile of a binomial distribution based on N samples (where N is the total number of neurons recorded at that age), and a 1.49% success probability.

### Graphical presentation of data
All data are presented as boxplots, unless the full distribution is also shown elsewhere as a histogram. Boxplots were generated using the Matlab (The MathWorks Inc, USA) function 'boxplot'. In all figures except Fig. 6, boxplots show the data median (central line), 25th and 75th percentiles (upper and lower box extremes), limits of the data excluding outliers (whiskers; outliers defined as lying more than 1.5 times the inter-quartile range beyond the upper or lower quartile), and 95 percent confidence interval for the median (notch in box; defined as $q2 - 1.57(q3 - q1)/\sqrt{(n)}$ and $q2 + 1.57(q3 - q1)/\sqrt{(n)}$, where $q2$ is the median, $q1$ and $q3$ are the 25th and 75th percentiles, respectively, and $n$ is the number of observations). In Fig. 6, due the compact nature of the plots, no notches are shown, and the median is shown by a circular target symbol.

### Assessing spatial tuning and stability of BVCs
The spatial specificity of neuron firing was assessed using Spatial Information, defined as the mutual information $I(R|X)$ between firing rate $R$ and location $X$:

$$I(R|X) \approx \sum_i p(x_i) f(x_i) \log_2 \left( \frac{f(x_i)}{F} \right) \qquad (4)$$

where $p(x_i)$ is the probability for the animal being at location $x_i$, $f(x_i)$ is the firing rate observed at $x_i$, and $F$ is the overall firing rate of the cell (Skaggs et al., 1996). $I(R|X)$ was then divided by the mean firing rate of the cell, giving an estimate in bits/spike. The spatial stability of BVCs was assessed using the Pearson's-r correlation of rate maps from

temporally adjacent square open field trials. Overall age trend in Spatial Information and Stability were tested using 1-way ANOVA (between-subjects factor age), and post-hoc pairwise comparisons using Tukey's HSD.

### Statistical testing of d, Φ, and $\sigma_0$ distributions
For each BVC, the $d$, $\Phi$, and $\sigma_0$ values were defined as those of the best-fitting model, in the open field trial with the highest fit-maximised Pearson's-r. Overall age trends in median $d$, and $\sigma_0$ were tested using a Kruskal-Wallis test (between-subjects factor age). To quantify the observed clustering of $\Phi$ tunings at the cardinal compass points, a Rayleigh test for unidirectional departure from uniformity was performed on $\Phi$ angle data that had first been quadrupled, and then wrapped onto the interval 0–360°. This test is well-suited to detect multimodal departure from uniformity, in cases where a clear hypothesis predicting $n$-fold circular symmetry of the modes exists[72]. Overall development trends in the proportion of BVCs with $\Phi$ tunings oriented to a wall (defined as cardinal compass points ±12°) was tested using a $\chi^2$ test. Differences in the proportion of wall-oriented $\Phi$ between long- and short-distance tuned BVCs (defined as $d > 10$ cm and $d \leq 10$ cm, respectively) were tested using $\chi^2$ test, post-hoc pairwise comparisons were performed using a $z$-test for proportions. The distribution of $\Phi$ tunings in the circular arena was defined as that of the model that best fit the circular arena neural rate map, and distributions were tested using the Rayleigh test on quadrupled, wrapped data. The mean rotation of BVC ensembles, between the square and circular arenas, was defined as the circular mean of the signed differences in $\Phi$ between arenas. BVCs with $d > 25$ cm were excluded from this estimate, as their rotations could be ambiguous across a 180° symmetry. The circular dispersion of rotations around ensemble means was assessed using the Kappa test of circular concentration, on the population of the differences between $\Phi$ rotation for each BVC, and the respective ensemble mean rotation. To obtain a reliable estimate of mean rotation, only ensembles with ≥5 simultaneously recorded BVCs were included in the analysis: in the following test for 4-fold symmetric clustering at the population level, all developing rats were analysed together, to compensate for the reduced number of BVCs resulting from this restriction. Analysis of changes in 4-fold symmetrical clustering within ensembles used all BVCs and only ensembles with ≥5 simultaneously recorded BVCs.

### BVC responses to inserted barriers
First, for each BVC the appropriate barrier orientation for assessing responses was determined on the basis of the directional tuning: BVCs with $\Phi$ tunings between 60° and 120° or between 240° and 300° (where 0° is East) were defined as north-south tuned BVCs, and responses were assessed using an east-west oriented barrier. BVCs with $\Phi$ tunings ≤30°, between 150° and 210° or ≥330° were defined as east-west tuned BVCs, and responses were assessed using a north-south oriented barrier. Those BVC whose $\Phi$ tunings fell outside these classifications were not considered as unambiguously north-south or east-west oriented and were excluded from the barrier analysis. A BVC was only included in the analysis if a trial with the appropriate barrier orientation had been conducted in that session, and the $d$ tuning of the BVC was <15 cm. Following this, BVC responses to barriers were assayed using a method similar to that described in ref. 30. Briefly, the changes in firing rate were measured in two zones defined relative to barrier position: each zone was as long as the barrier in the dimension parallel to the barrier, and extended from the barrier, to 12.5 cm away from the barrier, in the orthogonal dimension. For each BVC, the two zones were designated distal and proximal on the basis of the BVC's directional tuning, with the distal zone being on the opposite side of the barrier to the direction of $\Phi$. The absolute rate barrier response was then defined as the change in the summed values of rate map bins within each zone, between the barrier trial and the preceding open

field square. The normalised response was defined as the absolute response, divided by the sum of the summed rates in the barrier and open field trials. Overall age trends in barrier response were tested using a mixed design ANOVA, including age as a between-subjects factor and zone (distal versus proximal) as a within-subjects factor. Post-hoc pairwise comparisons were conducted using Simple Main Effects.

## Comparison of subiculum and mEC neural data

The mEC dataset analysed here was previously described in ref. 30. Data was shared in the form of position tracking records and spike times of isolated single units (the same set as described in the original study). All further analysis, including construction of rate maps, fitting of BVCs, and determining fit significance using shuffled data, was performed as described for subiculum data, above. Neurons were defined as border cells using the same method as in ref. 30, namely, both the Border Score[23] and the Spatial Information (see above) of a neuron's rate map needed to exceed the 95th percentiles of populations of Border and Spatial Information scores derived from age-matched, spatially-shuffled data. The Spatial Information, stability and BVC tuning properties of both BVCs and Border Cells identified in the mEC were analysed as for subiculum BVCs, as described above.

## Sub-sampling of data

Control analyses were performed to test whether developmental changes in path length per trial, running speed, mean rate or cluster stability contributed to results reported in Fig. 2, and whether lower BVC $r_{(max)}$ in the circle than the square contributed to the results reported in Fig. 4. The details of these are as follows:

*Path length* Path length was defined as the integrated sum of all position offsets in a recording trial, measured at a 50 Hz sample rate. Path lengths were equalised by discarding all data from the times at which rats had travelled 65 m in a trial onwards. Matched shuffled data was produced by shifting spike times relative to path after the discarding of data. BVC classification was performed as described above, using sub-sampled real and shuffled data. See Supplementary Fig. 2.

*Running Speed* Running speed was quantified using trial median speeds, after excluding immobility (defined as speed <2.5 cm/s). Speeds were equalised by discarding position samples at either the high or low ends of the speed distribution, such that all trial median speeds became 7.57 cm/s. Matched shuffled data was produced by shifting spike times relative to path after the discarding of data. BVC classification was performed as described above, using matched, sub-sampled real and shuffled data. See Supplementary Fig. 2.

*Mean rate* Mean rates were matched over the full population of recorded neurons, before BVC classification, to control for the possibility that changes in mean rate biased which neurons were selected as BVCs. Rates were matched by discarding a set proportion $P$ of the spikes of every cell, the discarded spikes being chosen randomly. $P$ varied by age group, and $P$ for each age group, $P_{(AG)}$, was defined as $1 - (M_{(min)}/M_{(AG)})$ where $M_{(AG)}$ is the population median mean rate for the age group, and $M_{(min)}$ is the minimum population median mean rate over all age groups. The proportions $P$ were: P16-18, 0.0149; P19-21, 0.0435; P22-25, 0; Adult, 0.2846. Matched shuffled data was produced by shifting spike times relative to path after the discarding of data. BVC classification was performed as described above, using matched, sub-sampled real and shuffled data. See Supplementary Fig. 3.

*Cluster Stability* was quantified by calculating waveform change, defined as the absolute change in the peak voltage of the mean cluster spike waveform across two adjacent trials. The tetrode channel with the maximum amplitude spike was used to calculate waveform change. Waveform change was equalised by discarding all neurons with a waveform change of >4.5 μV. Cluster isolation was quantified using the

Isolation Distance and L-Ratio of clusters[73], calculated in 4-dimensional spike amplitude space.

*BVC $r_{(max)}$ in the circle* was equalised with that in the square separately for each age group, by sorting cells by their circle BVC $r_{(max)}$, and progressively discarding cells from the circle dataset until the age group population medians were matched across square and circle. The analyses reported in Fig. 4 were then performed with this sub-sampled dataset (see Supplementary Fig. 7).

## BVC firing modulation by head direction

Directional firing rate maps were constructed as follows: for each head direction angular bin (bin width = 6°), the number of spikes occurring while the animal occupied the bin was divided by the total dwell time in the bin, and the resulting rate values were smoothed with a boxcar kernel (window width = 30°). The degree of head direction modulation was defined as the length of the mean resultant vector (Rayleigh vector) of the binned, smoothed, rate map. The preferred firing direction of each cell was defined as the angle of the Rayleigh vector. The extent to which directional modulation could be explained by a combination of spatially-restricted firing in (2-dimensional) space, and uneven sampling of direction across space, was quantified by producing directional rate maps following the distributive hypothesis[74]. Distributive hypotheses maps are produced by assuming that a neuron has no intrinsic head direction modulation, and calculating the expected firing rate for each directional bin, on the basis of (1) the directions sampled and (2) the mean firing rates recorded, in every (2-dimensional) spatial rate map bin[74]. The directional modulation of distributive hypothesis rate maps was quantified using the Rayleigh vector length. To give an estimate of the extent of residual directional modulation, which could not be explained by positional sampling bias, the Rayleigh vector of the distributive hypothesis map was subtracted from that of the observed data map.

## BVC simulation

To test whether the four-fold symmetry in $\Phi$ tunings could arise from behavioural bias, BVCs were simulated using age-matched real position data and synthetic spike trains. To create synthetic spike trains, the activation of a specific BVC receptive field was determined for every location visited by the rat in a trial, at camera pixel resolution (2.5 mm), by evaluating equation (3), above. The number of spikes occurring during each 20 ms position sample was then determined by a random Poisson process, in which Poisson $\lambda$ was linearly proportional to the model BVC firing rate $f(x,y)$ at the currently occupied location, scaled by a fixed factor such that the total number of expected spikes in a trial matched a mean rate drawn at random from an age-matched population of real BVCs. Following generation of the synthetic spike train, data were binned, smoothed, and BVC best fit determined following the same procedures as those used for real data (see above). Simulated BVCs were only included in the simulated population if the best fit $r_{(max)}$ surpassed the shuffled-data thresholds applied to real data. The tuning parameters associated with the best fitting BVC could potentially, therefore, be different to those used to create the synthetic spike train. 50,000 simulated BVCs were created for each age group. For each simulated BVC, spike trains were based on $d$ and $\sigma_0$ values drawn from age-matched populations of real data. $\Phi$ values were drawn from either a flat distribution, or, as a positive control, age-matched populations of real data $\Phi$ values. To test the likelihood of finding four-fold symmetry in a population of $N$ BVCs, where $N$ is the number of real BVCs found in each age group, $N$ simulated BVCs were sampled at random (without replacement) from the full population 1000 times. For each resampled population, of $N$ simulated BVCs, the presence of fourfold symmetry was tested using the Rayleigh test on quadrupled, wrapped $\Phi$ values.

## Field shuffle

The field shuffle was an alternative method for generating a spatially-shuffled null population of BVC fits, to test the statistical significance of real data BVC fits. The procedure for producing field shuffle data was based around that of ref. 39, in summary, being to segment the rate map based on field position, then randomly rearrange the resulting segments. First, spatial field peaks (defined as local maxima) were found in the binned, smoothed rate map for each cell. A prominence measure was calculated for each peak, defined as the ratio between the summed rates of the 9 bins including and immediately adjacent to the peak bin, and the surrounding 16 bins immediately adjacent to these 9. To avoid over-segmenting the map, if more than 8 peaks were detected, only the highest eight prominence peaks were used. Following peak detection, the map was divided into polygons defined by the Voronoi diagram of the peaks. The polygons were then randomly rearranged: each was assigned a random position and orientation within the arena, in order of the mean rate of their contained bins, starting with the polygon with the highest mean rate. If a polygon's random position overlapped with that of another already placed, the overlapping bins from the later-placed (lower mean rate) polygon were assigned randomly to unfilled bins, after all polygons were placed. Rate bin values within the placed polygons/bins were unsmoothed: smoothing was performed only after the polygon/bin placing procedure was complete. BVCs were then fit to the resulting spatially randomised, smoothed map.

## Fitting of BVC and place cell models to subiculum and CA1 data

CA1 data was taken from a previously published dataset[29], age-matched to the age groups used in the current study. Model BVCs were fit to CA1 data using the same procedures as for Subiculum data, as described above. To compare BVC fits to an alternative model, both CA1 and subiculum data were fit with model Place Cells. Following[33] these were defined as regular 2-dimensional gaussian fields, whose peak could take any position in the environment, and whose field width could be one of four $\sigma$ values: $\sigma = 7, 9, 11$ and 13 cm. Model place cells were radially symmetrical, and model fields near the arena boundaries were clipped at the boundaries, but not otherwise distorted. If the $r_{(max)}$ derived from the best-fitting Place Cell was larger than that derived from the best fitting BVC, the neuron was classified as being better fit by a Place Cell model. The proportions of significantly fit neurons in CA1 and Subiculum were determined with respect to the number of spatially-modulated neurons (defined as those whose Spatial Information scores surpassed the 75th percentile of a population of scores derived from a spike-time shuffled data) in each area and at each age: this was done to correct for higher levels of spatial modulation in CA1 overall, at all ages[51].

## Burst index

Burst index was measured using two complementary methods. The inter-spike interval histogram (ISI) method defines burst tendency as the ratio between the number of spike intervals <8 ms, and all inter-spike intervals. The ISI method may under-estimate bursting, when mean rates are low[75]. The auto-correlogram (AC) method defines burst tendency as the ratio between the integrated area under the temporal spike train AC in the period 1–6 ms, and that in the period 9–20 ms. The AC method may over-estimate bursting, when mean rates are low

## Reporting summary

Further information on research design is available in the Nature Portfolio Reporting Summary linked to this article.

## Data availability

Subiculum and mEC data are freely available to download at https://rdr.ucl.ac.uk/articles/dataset/Subiculum_neuron_data_from_adult_and_developing_rats/24864732. The DOI for this dataset is: 10.5522/04/24864732.

## Code availability

Custom code is available to download at: https://github.com/WillsCacucciLab/BVCDevPublic.

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

## Acknowledgements
We thank Colin Lever for revision and very helpful discussion of the manuscript. We thank the following funders for supporting this research: Wellcome Trust (Senior Research Fellowship 220886/Z/20/Z to T.J.W., PhD Studentship to F.R.R. and BT), The Royal Society (University Research Fellowships UF150692 and UF100746 to T.J.W.), Medical Research Council, UK (MR/N026012/1 to T.J.W.), European Research Council (Starter Award DEVSPACE and Consolidator Award DEVMEM to F.C.), Biotechnology and Biological Sciences Research Council, UK (grants BB/I021221/1 and BB/R009872/1 to F.C.). This research was funded in whole, or in part, by the Wellcome Trust, grant number 220886/Z/20/Z. For the purpose of Open Access, the author has applied a CC BY public copyright licence to any Author Accepted Manuscript version arising from this submission.

## Author contributions
Conceptualisation, Thomas J. Wills and Francesca Cacucci; Investigation, Laurenz Muessig, Fabio Ribeiro Rodrigues; Formal Analysis & Visualisation, Laurenz Muessig, Thomas J. Wills; Writing—Original Draft, Thomas J. Wills, Francesca Cacucci; Writing—Review & Editing, Thomas J. Wills, Francesca Cacucci, Laurenz Muessig, Fabio Ribeiro Rodrigues, Neil Burgess, Caswell Barry, Edvard I. Moser, May-Britt Moser; Funding Acquisition, Thomas J. Wills, Francesca Cacucci; Resources, Benjamin W. Towse, Caswell Barry, Neil Burgess (BVC fitting algorithm), Tale L. Bjerknes, Edvard I. Moser, May-Britt Moser (previously published data contributing to Fig. 6); Supervision, Thomas J. Wills, Francesca Cacucci.

## Competing interests
The authors declare no competing interests.
