## [Peer Review File · Nature Communications]

REVIEWER COMMENTS

Reviewer #1 (Remarks to the Author):

Summary:

In this work, the authors examined the emergence and general properties of subiculum boundary vector cells (BVCs) in young and adult rats in square and circular environments. BVCs were categorized based on a model put forth by Hartley et al., 2000, which assumes BVCs fire according to Gaussian tuning to a wall direction and angle (i.e., a cell fires when a boundary is a certain direction and distance from the animal's head). The authors observe that BVCs became more tuned with age, as they were much more tuned in adult than P25 or younger rats. This contrasts with the early robust activity of border cells in mEC. Interestingly, in square environments, BVCs exhibited a directional preference aligned with one of the wall directions. This alignment disappeared in circle environments, suggesting that environmental geometry influences BVC tuning – a novel finding but also consistent with properties of other spatial cell types. Further, while BVCs in all animals showed some response to an inserted border, this response strengthened with age. BVCs in adult animals also showed an inhibitory field on the opposite side of the inserted wall. Overall, this work highlights several interesting and new features of BVCs and maps out their development.

Overall, I felt this study was interesting and well-written – it was quite easy to follow and clearly presented, which is always much appreciated! However, I do have a few comments – mostly revolving around verifying some of the claims regarding the change in directional preference and the change in BVCs with development. In addition, while clearly presented, some of the results I believe can be a bit better motivated or contextualized, and perhaps more discussion on the impact and interpretation of the results can be added. More details are given below.

Major comments:

1. As much of the paper is a characterization of BVC using a particular model, I think it would help the reader to describe, at least in a sentence or two in the main text, what the model is rather than only citing the paper. This is particularly evident in the first paragraph of the results, where aspects of the model are discussed but the model itself is never clearly defined (although it is in Hartley et al and in the methods).
2. The emergence of BVC tuning with age (Fig 2) is quite interesting, especially as it departs from border cell development in mEC (Fig 6). However, it seems possible that differences in behavior – e.g., lower occupancy/coverage across spatial bins at younger ages – could influence this result. I couldn't find where the coverage or other aspects of behavior (e.g. running speed) were reported across groups. This seems to me to be an important thing to report given the differences described. I would suggest that the

authors report this, and perhaps determine whether changes in BVC tuning correlate with changes in behavior/coverage.

3. The distribution of directional tunings in the square is very interesting, but I have a few questions about this result. First, I wonder if it is possible if this could stem from the fact that firing is typically near a border (small d as shown in Fig 3A), and animals typically traverse borders in one of four directions. Relatedly, it's not clear how strongly the directional aspect of modulation is – in other words, how much does this aspect of the model influence or predict spiking? For example, if the cells were more like border cells, or specifically border cells with no directional aspect to their tuning, rather than BVCs, would this type of directional tuning 'falsely' emerge through the analysis? If the authors think idea has merit, I think that it should be straightforward to simulate border cells (or cells that fire in the same location but with no directional tuning) and test this.

4. Overall in the manuscript, while the results are interesting, I felt in some places it was perhaps too descriptive - I wasn't entirely sure what to make of the result, or how this should update my internal model (or any theoretical model) of spatial navigation as it relies on BVC tuning. I think to make the impact clearer, the authors could clarify the consequence of the reported effect in a few places:

a. The authors report what seems like two fundamentally new features of BVCs – the potential remapping of directional tuning across different environmental geometries, and the inhibitory receptive fields on the opposite side of an inserted wall. Do these results have any impact on models that use BVCs, even old place cell models that rely on BVCs? Does this sort of re-mapping, or the inhibitory field, serve a purpose based on existing hypotheses of BVC function? While the authors list cases in which the effects in BVCs seem to match that of other spatial cell types, it becomes unclear if this is just a downstream effect that the authors are picking up on, or if this points to something more foundational.

b. Regarding the developmental side of this paper – is there any impact of having BVCs emerge later than border cells in mEC?

Minor comments:

1. I did not follow the discussion on ensembles (lines 169-173). Is the alternate hypothesis that small groups of BVCs (those that remap together) exhibit the 90-degree tuning?

2. Neuron should be 'neurons' in line 48.

3. Some of the BVC tuning looks a bit weak, especially those in which the firing is not along much of the border but seems to be in small fields. I suppose part of the point is to also capture putative BVCs that might emerge early in development. However, is it possible to replicate some of the most important results (perhaps the directional result in Fig 4, and the boundary tuning in Fig 5) with BVC cells that are more clearly tuned, perhaps taking a more aggressive threshold?

Reviewer #2 (Remarks to the Author):

In this manuscript, Muessig et al. characterize the emergence of boundary vector cells (BVC) during development in the subiculum and mEC, and report some new characteristics of BVCs. BVCs are detected by the fitting of BVC models to experimental data. In the subiculum, the prevalence and stability of BVC increase throughout development, but no significant change in terms of BVC angular tuning, field width and boundary-distance is observed. In mEC, BVCs show adult-like levels of prevalence and stability from early developmental stages, indicating that BVCs mature earlier in mEC than in the subiculum. In terms of new BVC characteristics, the authors observed a tendency of BVC angular tuning to be aligned with walls in a square arena, but not in a circular arena. Furthermore, they report an inhibition of BVC firing field on the proximal side of inserted barriers.

The manuscript is generally interesting, clearly written and well organized. I have one concern which is related to the method for detecting the BVCs. Some of the BVC examples are not visually convincing, which makes wonder if the detection method is very accurate. The significance of the model fitting is estimated based on a shuffled distribution obtained by shifting cells' spike time. Such shifting of spike time is expected to produce rate maps with reduced spatial modulation (which is appropriate to assess significance in spatial modulation). However, I wonder if a reduction in spatial modulation might not systematically generate a decrease in correlation between rate maps and models. Then, to what extent the current method is not simply reflecting cells' spatial modulation? Also, I am not sure that using all cells within a group to estimate a single threshold for the whole group is totally appropriate in this case, given the variability in cells' firing rates.

In regard of the above points, I think the authors need to demonstrate the reliability of their method. The authors might show for instance that the method is not categorizing hippocampal place cells as BVCs, and that in the inserted boundary experiments, BVC detection is specific to cells developing an extra firing field on the distal sides of inserted barriers (if this is a determining criteria for BVC).

Reviewer #3 (Remarks to the Author):

In this manuscript, the authors quantified various properties of the boundary vector cells (BVCs) in the subiculum and characterized the postnatal developments of BVCs. They found a delayed emergence of BVCs after weaning, while the boundary vector tuning strength and spatial stability increased gradually across age. They also reported that the directional tuning of BVCs was significantly aligned with the orientations of the arena walls in the square maze. However, in the circular maze, the BVCs lost their wall-oriented directional tuning and reached an angular symmetry, suggesting that the environmental geometry influenced the BVCs' boundary tuning properties. By incorporating the data of early mature

BVCs recorded from the medial entorhinal cortex(mEC), the authors noted again that the subicular BVCs mature slowly until the late developmental stage. Although the results are novel and supported by detailed quantification, some points must be addressed to complement their arguments.

Major comments

1) The recording sites in the subiculum are shown in Figure S1, but many black arrowheads indicating tetrode positions seem to be outside the target region. Some tetrodes didn't even reach the subiculum and ended at the corpus callosum (e.g., r123 and r375). The authors must clarify this mismatch. The authors should present the exact positions of tetrode tips from which the BVCs were recorded. Additionally, it is necessary to describe the basic firing properties of subicular neurons, including both BVCs and non-BVC neurons, to align their findings with the results reported in prior studies. It is also strongly recommended to provide graphs showing the changes in peak amplitude across time to verify recording stability and unit-isolation quality.

2) It would be ideal for the authors to present whether non-BVC, spatially tuned neurons in the subiculum also mature at a late developmental stage, compared to BVCs. If what the authors claim is correct, non-BVCs are also expected to develop slowly if the hippocampal formation develops sequentially along the tri-synaptic loop. However, if the BVCs have late maturation because they require more complex computation or circuitry, BVCs may develop later than spatially tuned neurons in the subiculum. Related to this issue, in Figure 3, the boundary vector fields near the arena walls had mature directional tuning already at a young age. In contrast, the fields far from the walls developed their directional tuning slowly. These findings suggest that other factors, such as the development of vision, could affect the late maturation of BVCs. These points need to be addressed.

3) The authors tested the hypothesis that BVCs are affected by environmental geometry, and it has been demonstrated that there is a difference in BVC directional tuning. However, some distortions were made since the distal visual cues surrounding the two arenas were different, which may put some limitations on the argument made by the authors. Specifically, there is a possibility that the different and distorted directional tuning in those two mazes was induced by different distal visual cues rather than boundary geometry. It has been reported that the difference in environmental complexity (i.e., a complex environment with rich visual cues and an environment in which one salient cue is present) can make hippocampal formation operate differently (Shin et al., Cell Reports, 2022). In Lever et al. (2009), although there was no quantification, BVCs' directional tuning was maintained constantly across various geometry conditions, which may be attributable to the salient visual cue fixed at a particular position. To exclude the possibility that the different visual configuration causes the alteration of BVC properties, the authors should present all the BVC characteristics measured in the circular maze other than the directional tuning, as shown in Figure 2.

Other comments

1) The authors should provide a more detailed explanation of the recording schedule for rats in the methods section. For example, they should explain whether the rats always experienced the square maze first and then underwent the circular maze next or whether the rats went through the familiarization sessions before neural recording. Also, I want to know if two consecutive recording trials were conducted for the circular maze. If so, the results of spatial tuning and stability across age groups for the circular maze should also be presented so that readers can compare them with the results of the square maze.

2) Figure 6 should include how the BVC Fit (r_{\max}) of BVCs and border score (BS) of border cells in the mEC vary with age and how they are statistically different across age groups, which will correspond to the graph of BVC Fit (r_{\max}) in Figure 2D.

3) In the legend of Figure 2, the descriptions of panels C and D are reversed.

4) In Figure 6B, there are some cells for which BVC Fit (r) values are not provided.

We thank all reviewers for their helpful and constructive comments. We hope they will agree that we have satisfactorily addressed all their key concerns, most notably relating to:

- 1- the specificity of the BVC detection method (both by modifying the method used, and showing in supplementary analyses that our results are robust to further refinements);
- 2- whether non-specific changes in behaviour or network maturation drive the observed developmental trends (by sub-sampling data to match non-specific factors, and showing developmental trends are unchanged)
- 3- ruling out alternative explanations for the differences in BVC firing between square and circle environments (by providing a full characterization of firing in the circular environment).

Please note that, in accordance with Nature Communications guidelines, all bar charts have been replaced with boxplots, unless the full distribution is already shown elsewhere.

Answers to specific points are below:

Reviewer #1 (Remarks to the Author):

Summary:

In this work, the authors examined the emergence and general properties of subiculum boundary vector cells (BVCs) in young and adult rats in square and circular environments. BVCs were categorized based on a model put forth by Hartley et al., 2000, which assumes BVCs fire according to Gaussian tuning to a wall direction and angle (i.e., a cell fires when a boundary is a certain direction and distance from the animal's head). The authors observe that BVCs became more tuned with age, as they were much more tuned in adult than P25 or younger rats. This contrasts with the early robust activity of border cells in mEC. Interestingly, in square environments, BVCs exhibited a directional preference aligned with one of the wall directions. This alignment disappeared in circle environments, suggesting that environmental geometry influences BVC tuning – a novel finding but also consistent with properties of other spatial cell types. Further, while BVCs in all animals showed some response to an inserted border, this response strengthened with age. BVCs in adult animals also showed an inhibitory field on the opposite side of the inserted wall. Overall, this work highlights several interesting and new features of BVCs and maps out their development.

Overall, I felt this study was interesting and well-written – it was quite easy to follow and clearly presented, which is always much appreciated! However, I do have a few comments – mostly revolving around verifying some of the claims regarding the change in directional preference and the change in BVCs with development. In addition, while clearly presented, some of the results I believe can be a bit better motivated or contextualized, and perhaps more discussion on the impact and interpretation of the results can be added. More details are given below:

Major comments

1. As much of the paper is a characterization of BVC using a particular model, I think it would help the reader to describe, at least in a sentence or two in the main text, what the model is rather than only citing the paper. This is particularly evident in the first paragraph of the results, where aspects of the model are discussed but the model itself is never clearly defined (although it is in Hartley et al and in the methods).

We thank the reviewer for this suggestion. We have added a brief description of the BVC model to the beginning of the Results section. See lines 93-96. We have also replaced Figure panel 1A with a cartoon schematic of model BVC receptive fields, and the effect of changing

the model parameters varied in the BVC fitting procedure. We hope this satisfies the reviewer's request.

2. The emergence of BVC tuning with age (Fig 2) is quite interesting, especially as it departs from border cell development in mEC (Fig 6). However, it seems possible that differences in behavior – e.g., lower occupancy/coverage across spatial bins at younger ages – could influence this result. I couldn't find where the coverage or other aspects of behavior (e.g. running speed) were reported across groups. This seems to me to be an important thing to report given the differences described. I would suggest that the authors report this, and perhaps determine whether changes in BVC tuning correlate with changes in behavior/coverage.

We agree that it is important to rule out changes in behaviour during development as a confounding factor in the development of spatially tuned firing. A threshold for coverage of the environment was already in use in the initial submission (data only used in trials where >80% of spatial bins were visited), details of which were omitted from the methods due to an oversight. This has now been added to the methods section, see lines 764-765.

Additionally, as requested, we have provided data relating to the development of speed/coverage (Supplementary Figure 2A). Total path length, and number of bins visited do significantly increase with age, and median speed also significantly changes with age (though speed does not monotonically increase, as previously reported, see Wills, Cacucci et al, 2010, Science). However, sub-sampling data to equalise path length per trial (which also removes a significant effect of age on bin coverage) has no effect on the age trends reported (Supplementary Figure 2B), the same being also true of sub-sampling data to match median speeds (Supplementary Figure 2D). See Methods line 923 for details of analysis used. We therefore feel confident that changes in behaviour do not contribute to the reported developmental changes in BVCs.

3. The distribution of directional tunings in the square is very interesting, but I have a few questions about this result. First, I wonder if it is possible if this could stem from the fact that firing is typically near a border (small d as shown in Fig 3A), and animals typically traverse borders in one of four directions. Relatedly, it's not clear how strongly the directional aspect of modulation is – in other words, how much does this aspect of the model influence or predict spiking? For example, if the cells were more like border cells, or specifically border cells with no directional aspect to their tuning, rather than BVCs, would this type of directional tuning 'falsely' emerge through the analysis? If the authors think idea has merit, I think that it should be straightforward to simulate border cells (or cells that fire in the same location but with no directional tuning) and test this.

There may be some confusion as to the interpretation of the results described in Figure 3E. It is likely that our wording has led the reviewer to think that these results related to head direction tuning of BVCs. If so, we apologise for the confusion: BVCs are not strongly modulated by head direction, and the (small) head direction tunings that can be observed show no four-fold symmetry, and furthermore are largely explained by uneven sampling of direction across 2-dimensional space. We have added supplemental figure (5A-C) to illustrate this, and a sentence to the Results section (lines 156-160), in the interest of clarity and to avoid confusion. The data shown in Figure 3E instead relate to the directional orientation (ϕ) of the BVC receptive field which is dissociable from the animal's current heading direction.

Nevertheless, we felt that the reviewer's idea had merit, when applied to the four-fold symmetry of BVC ϕ tunings seen in our data, i.e. do behavioural biases contribute to the

clustering of BVC phi tunings we observed in the square environment? We therefore implemented the reviewer's suggestion, testing the potential effect of behavioural biases on detected BVC orientation by generating a population of neuronal responses whose directional tuning is uniformly distributed across 360 degrees ("uniform" simulated data). Simulations were obtained using real (age-matched) position data and synthetic spike trains, the latter based on a random Poisson process scaled by the BVC function (see Methods, lines 979-1000). The resulting spike trains were binned and smoothed, and BVC tunings detected following the same procedures as for real data. If the 4-way clustering in directional tuning (phi) is determined by behavioural biases, we should observe a 4-way clustering in the directional tuning (phi) of the simulated data. The results of the simulation are shown in Supplementary Figure 5D, E. Simulated data do not display the four-way clustering of phi.

As a positive control, we also simulated data reflecting the directional tunings observed in real data ("non-uniform" simulated data). Similarly to real data, data generated with the "non-uniform" simulation procedure display the four-fold clustering aligned with environment walls (Supplementary Figure 5F, G).

Overall, therefore, we find no evidence of a biased distribution of directional turnings in the square environment being induced by the animals' behaviour. These control procedures add to the evidence that the clustered distribution of phi tunings observed is a true feature of the firing of subiculum BVCs.

4. Overall in the manuscript, while the results are interesting, I felt in some places it was perhaps too descriptive - I wasn't entirely sure what to make of the result, or how this should update my internal model (or any theoretical model) of spatial navigation as it relies on BVC tuning. I think to make the impact clearer, the authors could clarify the consequence of the reported effect in a few places:

We thank the reviewer for this suggestion: we have tried to make the functional impact of our findings clearer, throughout the discussion, rewriting the relevant sections where necessary. See lines 421-424; 428-433; 437-445; 459-462.

a. The authors report what seems like two fundamentally new features of BVCs – the potential remapping of directional tuning across different environmental geometries, and the inhibitory receptive fields on the opposite side of an inserted wall. Do these results have any impact on models that use BVCs, even old place cell models that rely on BVCs? Does this sort of re-mapping, or the inhibitory field, serve a purpose based on existing hypotheses of BVC function? [.]

Recent models of BVC function posit that these cells form an allocentric representation of space with respect to the boundaries of the current environment (Bicanski & Burgess, 2018). One key functional prediction of our results is that this representation should be distorted (and therefore less accurate) following transfer between two arenas of differing geometry. Interestingly, there is a rodent study in the literature that describes an effect consistent with this – platform location recall in a watermaze is disrupted following a switch from square to circle of vice versa (Bye et al, 2019, cited in manuscript). Clearly more work is needed to investigate the relationship between this behavioural effect and BVCs, but we can make a specific prediction, on the basis of our current results: that due to the nature of BVC receptive field distortion we observed, the behavioural effects of switching shape will be maximal at environmental corners (where the BVC representation changes between circle and square) and minimal/negligible in locations near the middle of walls (where the BVC representation

changes relatively little). We have added these considerations to the Discussion, lines 429-441.

The key functional implication of proximal-side receptive field inhibition is that it reflects a potential neural mechanism for attractor dynamics in the BVC network, which in turn provides a circuit-level explanation for the previously reported coherence of the BVC representation (i.e. simultaneously recorded BVCs do not remap, and if their phi tunings rotate, they do it rigidly and coherently - all together). Our data show that BVCs are inhibited at boundaries 180 degrees opposed to their 'canonical' field, by neurons that respond to the directionally opposed boundary (likely other BVCs, whether directly or indirectly). This putative connectivity between BVCs with opposed phi tunings could allow a self-sufficient continuous attractor to emerge within the BVC network, analogous to head-direction or grid cell continuous attractors, with important functional implications following from the existence of a coherent representation of 2-dimensional space, independent from that of grid cells.

These arguments are outlined in the paragraph at lines 442-453, in the discussion.

Lastly, an important question opened up by our results is how geometry determines BVC tuning (the relative importance of sensory input and motor behaviour patterns, which sensory modalities, etc). One important finding in this respect are subiculum corner cells, described in a *recent* BioRxiv manuscript (Sun et al, 2023) which are likely an important part of the neural circuits by which geometry influences neural representations and behaviour. This study is now briefly discussed at line 399.

[.] *While the authors list cases in which the effects in BVCs seem to match that of other spatial cell types, it becomes unclear if this is just a downstream effect that the authors are picking up on, or if this points to something more foundational.*

We argue that the effects of geometry on BVC tunings are unlikely to be downstream of other cell types. Regarding grid cells, BVC tuning distortion in the square is present at ages when spatially stable grid cell firing is yet to emerge, therefore it is clear that BVC receptive field distortion cannot be a downstream effect of grid cell 'shearing' in the square.

Regarding Egocentric Boundary Cells (EBCs), it is important to note that, unlike BVCs, EBCs do *not* change their receptive fields in response to boundary geometry. Instead, a four-fold symmetry in head-direction tuning emerges in the square, as a direct result of geometry constraining the orientations in which the animal can occupy a specific egocentric bearing to the wall. The relationship between EBC and BVC firing remains to be established, however, given the considerations above, we think that distortion of BVC firing in square environments is a conceptually distinct phenomenon from the four-way head direction tuning of EBCs.

We have added a paragraph, lines 412-425, in the Discussion section of the manuscript, to clarify these issues.

b. Regarding the developmental side of this paper – is there any impact of having BVCs emerge later than border cells in mEC?

A key prediction following from our results is that accurate spatial representations, and therefore accurate navigation, should emerge earlier for locations close to, rather than far from boundaries, due to the earlier emergence of border cells and short-range BVCs in mEC, which fire stably in close proximity to the border. To reflect this, we have added text to the discussion at lines 469-470.

Minor comments:

1. I did not follow the discussion on ensembles (lines 169-173). Is the alternate hypothesis that small groups of BVCs (those that remap together) exhibit the 90-degree tuning?

We apologise for any confusion here: the 'ensemble' in question is the ensemble of all BVCs simultaneously recorded from one rat, in one experimental session. The manuscript reports that all BVCs within these simultaneously recorded ensembles rotate coherently, when moving between circle and square. The rotation amounts may be different across rats/sessions, and the analyses in Figures 4E-F show that no 4-fold symmetry is present in phi tunings, even after correction for these different rotations. To avoid the confusion arising, we have clarified the wording of the results at the following points:

Lines 214-15: After 'Within each ensemble', insert '(comprising all simultaneously recorded BVCs in one recording session),'

Lines 225-6: Now reads: '.. Φ clustering is present in the circle within each simultaneously recorded BVC ensemble, but the orientation of these clusters is inconsistently aligned across experimental sessions, obscuring the clustering when all data are aggregated'.

2. Neuron should be 'neurons' in line 48.

Done

3. Some of the BVC tuning looks a bit weak, especially those in which the firing is not along much of the border but seems to be in small fields. I suppose part of the point is to also capture putative BVCs that might emerge early in development. However, is it possible to replicate some of the most important results (perhaps the directional result in Fig 4, and the boundary tuning in Fig 5) with BVC cells that are more clearly tuned, perhaps taking a more aggressive threshold.

We agree that the spatial firing of some of the BVC examples shown may be a less precise fit to the canonical BVC model, compared to (adult) examples described in previous work. However, as the reviewer correctly points out, it is important for the purposes of this study, with its developmental aspect, that we set our threshold for BVC definition inclusively enough that immature, developing BVCs are captured by our analysis. We have now added supplementary analyses describing modifications to our method (Supplementary Figures 9, 10) that later studies (perhaps those studying only adults) can, if necessary, use to capture only those BVCs which fit the model more stringently.

Specifically, in answer to this request and the request from Reviewer 2 (point 1), Supplementary Figure 9 describes the results of an alternative shuffling method ('field shuffle') which produces a higher classification threshold for BVCs, thereby reducing the size of the BVC population by approximately 40% (see comparison between Supplementary Figure 9B and 9D). Supplementary Figure 9F-M shows that the key results reported in this manuscript survive when using this higher threshold: we hope that this satisfies the reviewer's request.

Reviewer #2 (Remarks to the Author):

In this manuscript, Muessig et al. characterize the emergence of boundary vector cells (BVC) during development in the subiculum and mEC, and report some new characteristics of BVCs. BVCs are detected by the fitting of BVC models to experimental data. In the subiculum, the prevalence and stability of BVC increase throughout development, but no significant change

in terms of BVC angular tuning, field width and boundary-distance is observed. In mEC, BVCs show adult-like levels of prevalence and stability from early developmental stages, indicating that BVCs mature earlier in mEC than in the subiculum. In terms of new BVC characteristics, the authors observed a tendency of BVC angular tuning to be aligned with walls in a square arena, but not in a circular arena. Furthermore, they report an inhibition of BVC firing field on the proximal side of inserted barriers.

The manuscript is generally interesting, clearly written and well organized. I have one concern which is related to the method for detecting the BVCs. Some of the BVC examples are not visually convincing, which makes wonder if the detection method is very accurate. The significance of the model fitting is estimated based on a shuffled distribution obtained by shifting cells' spike time. Such shifting of spike time is expected to produce rate maps with reduced spatial modulation (which is appropriate to assess significance in spatial modulation). However, I wonder if a reduction in spatial modulation might not systematically generate a decrease in correlation between rate maps and models. Then, to what extent the current method is not simply reflecting cells' spatial modulation? Also, I am not sure that using all cells within a group to estimate a single threshold for the whole group is totally appropriate in this case, given the variability in cells' firing rates.

We thank the reviewer for these suggestions. Regarding the quality of BVC detection, we accept the reviewer's point that some example BVCs do not match the canonical model as precisely as some previously reported examples, from adult studies. However, as pointed out by Reviewer 1 (minor point 3), given the developmental aspect of our study, it is important that our threshold for BVC classification capture some of the more immature examples of BVC firing. The examples chosen in Figure 1C are intended to show a *representative range* of 'quality' (i.e. $r(\max)$) of BVCs captured by our classification procedure, rather than the best examples (including some examples which have only just surpassed the classification threshold). To make this clearer, we have modified the figure slightly, and removed the bottom line (a 'non-BVC') to make this clearer for the reader.

Notwithstanding this, the reviewer raises some important questions regarding the method for generating a classification threshold, arguing that (1) shuffled-data thresholds which are specific to individual cells/trials ('rate map shuffle') might be more appropriate than one based on an aggregated, age-matched population ('population shuffle') and (2) a shuffling method that preserves some of the spatial structure of firing ('field shuffle') could be more appropriate than one which shuffles spike times only. We investigated the outcomes of both of these methods: the results are presented in Supplementary Figure 9.

With regards to point 1, we found that 'rate map' shuffling produces a range of thresholds, many of which are lower than the currently used 'population' threshold (compare grey histograms with black dashed lines in Supplementary Figure 9A). Thus, a straightforward switch to a rate map threshold would not necessarily lead to a more stringent selection of BVCs. In the revised manuscript, we therefore use a conservative combination of both methods: cells are defined as BVCs only if their BVC-fit scores simultaneously exceed both the 'population' and 'rate map' thresholds (see revised Methods lines 805-816).

As to point 2, we tested the effects of a 'field shuffle', in which the spatial firing fields of the rate map are randomly rearranged, following a protocol similar to that of Krupic, Burgess & O'Keefe, Science, 2012. (see Methods lines 1002-1019 for full details). The field shuffle produces higher $r(\max)$ thresholds (for a direct comparison see Supplementary Figure 9A and 9C) and selects a sub-set of approximately 60% of BVCs classified by the temporal shuffle (see 9E). We felt this thresholding method too stringent to be used as the principal classification method for a developmental study. However, we now include in Supplementary

Figure 9 (9F-M) a reanalysis of our data using this higher threshold and find that key results are unaffected (with the exception of ensemble analyses of developing phi tunings in the circle, as there are no ensembles with ≥ 5 BVCs in these age groups, following application of the higher threshold). See also Results section, lines 276-297 and Discussion section, lines 374-386.

We hope this satisfies the reviewer's request.

In regard of the above points, I think the authors need to demonstrate the reliability of their method. The authors might show for instance that the method is not categorizing hippocampal place cells as BVCs, [...]

We thank the reviewer for their suggestion of testing the specificity of the BVC model fit method. As suggested, we applied our BVC detection procedure to age matched CA1 data (both developmental and adult, see Methods). We found that there were significantly fewer BVCs in CA1 than subiculum (in adults) but that there were nevertheless CA1 neurons classified as BVCs by our method. This was more evident in developing pups than in adults, and when using the spike-time shuffle rather than the field shuffle (Supplementary Figure 10A).

We argue that this may not simply be due to poor specificity of our BVC detection method: boundaries are known to be an important determinant of place field shape (Muller & Kubie, 1987, O'Keefe & Burgess, 1996, Barry et al, 2006), and a sub-set of place cells that take on BVC-like form may be expected in a walled arena. Examples are shown in Supplementary Figure 10B – many CA1 'BVCs' display place fields which are elongated along a boundary.

To help add more specificity to our detection method, and exclude 'BVC-like' place fields, we also fit all data with model 'Place Fields' (defined as radially symmetric 2-dimensional Gaussian fields of varying positions and widths, see Methods lines 1021-1035) and excluded from the BVC dataset any cell whose Place Cell $r(\max)$ was greater than its BVC $r(\max)$. Following the addition of this criterion ("Gaussian field removal"), we observed a large (approx. 67%) reduction in the proportion of CA1 cells classified as BVCs, most notably in adults and post-weaning pups, for both the spike-time and field shuffle methods (Supplementary Figure 10C). Some Subiculum BVCs are also excluded when using this criterion, however, this did not alter any of the key results reported in this study (Supplementary Figure 10D-K for spike time shuffle, Supplementary Figure 10L-S for field shuffle).

Even after excluding neurons better fit by Place Cells than BVCs, there remained similar numbers of BVCs across CA1 and Subiculum in pre-weanling animals: we argue that this does not reflect poor specificity in our classification methods, but rather, is consistent with the previously reported requirement for boundary-proximity for CA1 place fields to be stable, at these ages (Muessig et al, 2015, Neuron).

In light of the results of these control analyses, and the fact that the current study is focused on the characterisation of immature BVCs, we have opted to retain a more inclusive classification method of BVC responses (see Discussion, lines 374-386).

[...] and that in the inserted boundary experiments, BVC detection is specific to cells developing an extra firing field on the distal sides of inserted barriers (if this is a determining criteria for BVC).

We did not use responsiveness to barriers as a defining criterion for BVC classification, for two reasons: (1) we wanted to be able to use barrier responsiveness as a *confirmation* of

whether BVC fitting in the open field successfully identified BVCs, and (2) we did not want to assume that barrier responses emerged at the same time as open-field BVC responses during development (as responses to the relatively novel barrier may take longer to emerge than those to the relatively familiar arena walls, for example). However, following the reviewer's request, we now report the proportion of BVCs which showed an increase in rate on the distal barrier side, following barrier insertion (see lines 245-6), to give a further indication (in addition to the data in Figure 5) of how successful our method is in selecting cells which double their fields in response to an inserted boundary.

Reviewer #3 (Remarks to the Author):

In this manuscript, the authors quantified various properties of the boundary vector cells (BVCs) in the subiculum and characterized the postnatal developments of BVCs. They found a delayed emergence of BVCs after weaning, while the boundary vector tuning strength and spatial stability increased gradually across age. They also reported that the directional tuning of BVCs was significantly aligned with the orientations of the arena walls in the square maze. However, in the circular maze, the BVCs lost their wall-oriented directional tuning and reached an angular symmetry, suggesting that the environmental geometry influenced the BVCs' boundary tuning properties. By incorporating the data of early mature BVCs recorded from the medial entorhinal cortex (mEC), the authors noted again that the subicular BVCs mature slowly until the late developmental stage. Although the results are novel and supported by detailed quantification, some points must be addressed to complement their arguments.

Major comments:

1) The recording sites in the subiculum are shown in Figure S1, but many black arrowheads indicating tetrode positions seem to be outside the target region. Some tetrodes didn't even reach the subiculum and ended at the corpus callosum (e.g., r123 and r375). The authors must clarify this mismatch. The authors should present the exact positions of tetrode tips from which the BVCs were recorded.

We apologise for the confusion caused by our labelling system for Supplementary figure 1. The black arrows in Figure S1 were not intended to show final tetrode tip position in the subiculum, but rather indicate tetrode tracks, more generally. We apologise for this poor presentation, and have now redrawn the arrows (now red for clarity) such that they point at tetrode track ends. We have also changed and improved the outlining of the subiculum. Additionally, following the reviewer's request, we have now carefully selected histology images showing the deepest extent of tetrode penetration for each rat. Images for rats r123, r156, r272, r124, r157, r137, r217, r247, r316, r346, r372 and r2138 have changed. (The labelling for r373 and r375 was swapped in the original manuscript by mistake).

Additionally, it is necessary to describe the basic firing properties of subicular neurons, including both BVCs and non-BVC neurons, to align their findings with the results reported in prior studies.

We thank the reviewer for this suggestion and agree that addition of this information is important to aid comparison across studies. Following the reviewer's suggestion, we now report mean firing rate (Supplementary figure 3) and burst index (Supplementary figure 12) for both BVCs and non-BVC neurons.

The mean rate (of all subiculum neurons, and of BVCs specifically) is shown in Supplementary Figure 3A. In order to control for the notable increases in mean firing rate during development,

we sub-sampled the data (removed spikes at random) and recalculated BVC spatial tuning and stability. As can be seen in Supplementary Figure 3B-C, the developmental trends reported in Figure 2 were not affected by equalising mean rates across development.

Burst firing tendency is characterised in Supplementary Figure 12D-E. As noted in relation to Point (2), below, burst firing mirrors mean rate in being slow to mature.

It is also strongly recommended to provide graphs showing the changes in peak amplitude across time to verify recording stability and unit-isolation quality.

We have added the requested information, in Supplementary Figure 4. There were no significant changes in cluster isolation (as quantified by L-Ratio and Isolation Distance) across age groups, ruling out cluster isolation as a confounding variable in our results. Waveform instability (as measured by peak amplitude change) was slightly but significantly higher in young rats. However, sub-sampling the data so as to match waveform change across ages (Supplementary Figure 4C) did not affect any of the reported developmental trends (comparison between Supplementary Figure 4D and Figure 2), ruling out recording instability as a potential cause of our results (see added text on lines 133-134).

2) It would be ideal for the authors to present whether non-BVC, spatially tuned neurons in the subiculum also mature at a late developmental stage, compared to BVCs. If what the authors claim is correct, non-BVCs are also expected to develop slowly if the hippocampal formation develops sequentially along the tri-synaptic loop. However, if the BVCs have late maturation because they require more complex computation or circuitry, BVCs may develop later than spatially tuned neurons in the subiculum.

We thank the reviewer for this suggestion. We tested developmental trends in Spatial Information and spatial firing stability in spatially-modulated subiculum cells which were not BVCs (see Methods lines 822-25 for classification criteria): the results are shown in Supplementary Figure 12A-C (and described in the Results section, lines 343-47). For all measures, there were strong and significant age trends, suggesting that the spatial firing of the entire population of spatially selective neurons in subiculum matures slowly, and that this is not a peculiarity of BVCs.

Furthermore, we note that both mean rate (Supplementary Figure 3) and burst firing (Supplementary Figure 12) show strong developmental increases over the age range studied, suggesting that basic physiological parameters of the subiculum network matures late (see note added to Results section at lines 348-51).

Related to this issue, in Figure 3, the boundary vector fields near the arena walls had mature directional tuning already at a young age. In contrast, the fields far from the walls developed their directional tuning slowly. These findings suggest that other factors, such as the development of vision, could affect the late maturation of BVCs. These points need to be addressed.

The reviewer makes a very interesting observation, i.e. that BVCs with long distance tunings may be more vulnerable to disruption from reduced visual acuity in early development, as their firing occurs further from visual landmark cues. They may therefore be more unstable, and this may cause the lack of four-fold symmetry in long-range BVC phi tunings, specifically at P16-18 (Figure 3H).

To test this hypothesis, we compared the development of spatial tuning and stability for BVCs with long (>10cm) and short-range d tunings. These results are shown in Supplementary Figure 6 (text in Result section, lines 180-193). For all measures tested, there was no specific age-related degradation in spatial firing or stability, specifically in young animals (i.e., no significant interaction terms in Age* d Tuning Distance ANOVA). The spatial information and BVC fit $r(\max)$ were significantly lower for long-range d BVCs, but this was constant across ages. There was no overall effect of age or d tuning on intra-trial stability. There was a non-significant trend towards lower inter-trial stability (stability across consecutive trials) for long-range d BVCs, specifically in the P16-18 age group: however, instability across trials should not affect the distribution of phi tunings in one individual trial.

Overall, we hope that the reviewer agrees with us that these results appear to rule out the hypothesis that long-range BVCs lack four-fold symmetry in their phi tuning due to instability (such as that caused by poorer visual acuity).

3) The authors tested the hypothesis that BVCs are affected by environmental geometry, and it has been demonstrated that there is a difference in BVC directional tuning. However, some distortions were made since the distal visual cues surrounding the two arenas were different, which may put some limitations on the argument made by the authors. Specifically, there is a possibility that the different and distorted directional tuning in those two mazes was induced by different distal visual cues rather than boundary geometry. It has been reported that the difference in environmental complexity (i.e., a complex environment with rich visual cues and an environment in which one salient cue is present) can make hippocampal formation operate differently (Shin et al., Cell Reports, 2022). In Lever et al. (2009), although there was no quantification, BVCs' directional tuning was maintained constantly across various geometry conditions, which may be attributable to the salient visual cue fixed at a particular position. To exclude the possibility that the different visual configuration causes the alteration of BVC properties, the authors should present all the BVC characteristics measured in the circular maze other than the directional tuning, as shown in Figure 2.

We thank the reviewer for this suggestion – as requested, spatial tuning, stability and receptive fields of BVCs in the circle are now fully characterised, in Supplementary Figure 7A-B. Overall, there are no significant differences between BVC spatial firing in square and circle: this included spatial information, intra-trial stability, distance tuning and sigma zero tuning (As only one trial was recorded in the circular arena on any given recording session, inter-trial stability could not be calculated). There was one exception to this pattern: the $r(\max)$ of the best fitting model BVC was significantly lower in the circle than in the square (consistently across age groups). This raises the important possibility that the lack of four-fold symmetry in phi tunings in the circle, could be due to the presence of poorer fitting BVCs (which might not align with the square cardinal wall orientations).

To test whether the distribution of phi tunings in the circle is affected by lower BVC $r(\max)$, we sub-sampled the BVC population to match median $r(\max)$ across square and circle, in each age group (Supplementary Figure 7C). Following $r(\max)$ matching, there were no significant differences across spatial firing properties, between square and circle (Supplementary Figure 7C-D). Crucially, the key results reported in Figure 4 were unaffected by $r(\max)$ matching (Supplementary Figure 7E-G): a uniform distribution of phi tunings was observed in the circle. The difference between phi tunings across square and circle appears specific to this characteristic of BVCs, therefore, and not related to any other property of BVC firing (text in Results section, lines 204-211).

Other comments

1) The authors should provide a more detailed explanation of the recording schedule for rats in the methods section. For example, they should explain whether the rats always experienced the square maze first and then underwent the circular maze next or whether the rats went through the familiarization sessions before neural recording. Also, I want to know if two consecutive recording trials were conducted for the circular maze. If so, the results of spatial tuning and stability across age groups for the circular maze should also be presented so that readers can compare them with the results of the square maze.

This information has been added to the Methods: see lines 726-30. Unfortunately, due to the limits of developing rat stamina, only one circle trial was run in each experimental session, therefore inter-trial stability in the circle could not be assessed.

2) Figure 6 should include how the BVC Fit (r_{max}) of BVCs and border score (BS) of border cells in the mEC vary with age and how they are statistically different across age groups, which will correspond to the graph of BVC Fit (r_{max}) in Figure 2D.

A new panel has been added to Figure 6 (panel F) showing the development of BVC model fit (r_{max}) for BVCs and Border Score for border cells. The associated statistics are now shown in the Results section (lines 325 and 331). There is no significant age trend for either measure, further supporting the argument that border-responsive cells in mEC do not display significant maturation over the period P16 to Adulthood.

3) In the legend of Figure 2, the descriptions of panels C and D are reversed

This is now corrected.

4) In Figure 6B, there are some cells for which BVC Fit (r) values are not provided.

Apologies for this error, the values have now been added.

REVIEWERS' COMMENTS

Reviewer #1 (Remarks to the Author):

The authors did a lovely job addressing my concerns. I believe that the additional analysis controls, and the additional statements addressing the impact of their results, improve the manuscript.

Reviewer #2 (Remarks to the Author):

The authors have addressed my concern by testing an additional shuffling method preserving cells' spatial information and by comparing the BVC model fittings with place field model fittings. I found the later aspect particularly elegant and convincing as the shuffling/fitting method (which might not seem necessarily unbiased) is equally applied to both place cell and BVC models. I have no further comments.

Reviewer #3 (Remarks to the Author):

Overall, the authors have successfully addressed my primary concerns for the previous manuscript version. They have included the developmental change of the Non-BVC (all neurons vs. spatially modulated neurons), which turned out not very different from the BCV. It would still be ideal to show some non-BVC examples (missing in the current manuscript).

We are happy to note that all reviewers were fully satisfied with our replies to their comments. We thank all reviewers for their time and the very helpful and constructive feedback, which has resulted in an improved manuscript.

We include the full reviewers comments, and any further replies below.

Reviewer #1 (Remarks to the Author):

The authors did a lovely job addressing my concerns. I believe that the additional analysis controls, and the additional statements addressing the impact of their results, improve the manuscript.

No further comments to address.

Reviewer #2 (Remarks to the Author):

The authors have addressed my concern by testing an additional shuffling method preserving cells' spatial information and by comparing the BVC model fittings with place field model fittings. I found the later aspect particularly elegant and convincing as the shuffling/fitting method (which might not seem necessarily unbiased) is equally applied to both place cell and BVC models. I have no further comments.

No further comments to address.

Reviewer #3 (Remarks to the Author):

Overall, the authors have successfully addressed my primary concerns for the previous manuscript version. They have included the developmental change of the Non-BVC (all neurons vs. spatially modulated neurons), which turned out not very different from the BCV. It would still be ideal to show some non-BVC examples (missing in the current manuscript).

As requested, we have reinstated the example non-BVC in figure 1.